# Transforming the German Food System: How to Make Start-Ups Great!

Kathrin Ludwig [1,2,3], Adriano Profeta [1] , Alexander Märdian [2], Clemens Hollah [1], Maud Helene Schmiedeknecht [3,*] and Volker Heinz [1]

1   DIL Deutsches Institut für Lebensmitteltechnik e.V.—German Institute of Food Technologies, Prof.-von-Klitzing-Straße 7, 49610 Quakenbrück, Germany; k.ludwig@dil-ev.de (K.L.); a.profeta@dil-ev.de (A.P.); c.hollah@dil-ev.de (C.H.); v.heinz@dil-ev.de (V.H.)
2   DIL Innovation Hub, Prof.-von-Klitzing-Straße 7, 49610 Quakenbrück, Germany; a.maerdian@dil-tec.de
3   ESB Business School, Reutlingen University, Alteburgstraße 150, 72762 Reutlingen, Germany
*   Correspondence: maud.schmiedeknecht@reutlingen-university.de

**Abstract:** The food system represents a key industry for Europe and Germany in particular. However, it is also the single most significant contributor to climate and environmental change. A food system transformation is necessary to overcome the system's major and constantly increasing challenges in the upcoming decades. One possible facilitator for this transformation are radical and disruptive innovations that start-ups develop. There are many challenges for start-ups in general and food start-ups in particular. Various support opportunities and resources are crucial to ensure the success of food start-ups. One aim of this study is to identify how the success of start-ups in the food system can be supported and further strengthened by actors in the innovation ecosystem in Germany. There is still room for improvement and collaboration toward a thriving innovation ecosystem. A successful innovation ecosystem is characterised by a well-organised, collaborative, and supportive environment with a vivid exchange between the members in the ecosystem. The interviewees confirmed this, and although the different actors are already cooperating, there is still room for improvement. The most common recommendation for improving cooperation is learning from other countries and bringing the best to Germany.

**Keywords:** start-up; innovation ecosystem; food system; transformation; Germany; food science; entrepreneurship; food technology; sustainable development goals; learning from other countries

## 1. Introduction

The food system faces significant challenges, from meeting the growing demand for food to reducing its environmental impact [1,2]. Furthermore, these challenges will continue to rise, and the food system requires a transformation [3–8]. Even though the literature around the topics of start-ups and innovation (eco-)systems are numerous, specific literature for food start-ups is limited. Hence, this was seen as a starting point for the present research.

In this paper we analyse drivers and barriers for aligning the German food sector based on innovations into a more sustainable direction. This contribution builds on the question of to what extent transformative innovations of start-ups in the food system can be supported by actors in the innovation ecosystem. For this purpose, these actors were interviewed with a focus on start-ups and persons engaged in innovation hubs for fostering innovations in the food sector. Although there are many non-industry-specific studies and country comparisons concerning start-ups, the current research concerning agriculture and food start-ups is still limited [9]. Nonetheless, start-ups in the food sector have experienced some momentum and major investments in the last few years [10].

Although many factors can positively impact the food system [11], this paper focuses on innovation as a possible facilitator to transform the food system. In this context, start-up

companies are a "crucial driver for innovation, economic development and renewal" [12] as today's start-ups are tomorrow's mid-sized sector [13]. Additionally, start-ups are often responsible for disruptive innovations.

This paper deals with innovation from the perspective of start-ups in Germany which can change or transform the current food system. It aims to identify how the success of start-ups in the food system can be supported and increased by current stakeholders in the innovation ecosystem in Germany. Therefore, the question is broken down into four aspects: (1) the challenges of food start-ups with transformative innovations, (2) resources available to support the success of food start-ups, (3) identifying current actors in the innovation ecosystem and what they are doing to drive start-up success, and (4) what measures need to be expanded or established to improve the innovation ecosystem for food start-ups in Germany.

In this study, the focus was on Germany because this country is one of the leading food export nations. The export business contributes to the industry turnover and is the growth engine for the food industry. Every third euro is earned abroad today. In 2020, food worth EUR 61.6 billion was exported [14]. Therefore, German food producers must be very competitive in order to maintain their market power. At the same time the production causes an enormous environmental impact. Despite the fact that Germany is labeled an innovative country by the Global Innovation Index [15], the country underperforms when it comes to entrepreneurial activities [16]. Compared to countries with a similarly high income, Germany is ranked 28 out of 33 countries [16]. Nonetheless, the establishment of new businesses and start-ups is essential for the economic growth and ability to innovate in a given country [17], as innovations are most likely to emerge from start-ups [18].

The result of this work is an overview of challenges, actors, and resources as well as practical recommendations for improving the current innovation ecosystem and thus, the potential success of start-ups in the food system.

In the following section an overview about current challenges and starting points for food system innovations is given. Subsequently, the research method and data collection are described. The qualitative findings are presented in chapter 4 and conclusions are drawn in chapter 5.

## 2. Overview about the Current Challenges and Starting Points for Food System Innovations

### 2.1. Current Challenges

The current food system is not sustainable and the way food is produced, bought, and consumed needs to be changed in the future [19]. Thus, the system is in need of a transformation [3–8]. In the past, the food system evolved to meet the growing demand for food worldwide [1]. However, this development was accompanied by severe and persistent problems. The system faces diverse sets of serious and interrelated challenges (see Table 1).

**Table 1.** Challenges of food systems.

| Category | Challenge | Evidence |
|---|---|---|
| Natural resources | Greenhouse gas emissions | Food system is responsible for 21–37% of total greenhouse gas emissions worldwide [20] |
| | Biodiversity loss | 16.5% of vertebrates and pollinators threatened with extinction [21] |
| | Water scarcity and pollution | Agriculture's share of water usage accounts for 70% of global freshwater and is a major contributor to water pollution [20] |
| | Food loss and food waste | 1.3 B tons yearly [22] |
| Demographics & Health | Population growth | Expected growth till 2050 to 9.7 B people from 7.8 B people in 2021 [23] |
| | Undernourishment | 8.9% of the world population is undernourished, i.e., 688 M in 2019 [20] |
| | Adult obesity | Over 13.1% in 2016 [20] |
| | Childhood overweightness and obesity | 5.6% or 38.3 M children under five were overweight in 2019 [20] |

Source: own table based on den Boer [1].

The food sector is highly dependent on natural resources [24]. It is the largest sector emitting greenhouse gases, has the largest influence on biodiversity loss, and is a massive

consumer and polluter of freshwater [6]. Additionally, 1.3 billion tons of food are wasted each year [22]. Furthermore, there is an unfair distribution of food as, on the one hand, 8.9% of the world population, amounting to 688 million people, are undernourished and, on the other hand, over 13.1% of adults and 5.6% or 38.3 million children under five are overweight [20].

These challenges are amplified by trends such as demographic changes, e.g., a growing world population, climate change, urbanisation and consumerism [25]. Thus, a growing population is not the main driver of demand but a combination of factors, including increasing per capita incomes, changing dietary preferences or cash-cropping [3]. Finding reasonable solutions to these challenges is critical to achieving the targets of the Paris Climate Agreement and the United Nation's Sustainable Development Goals [1]. There is a need for sufficient, safe, healthy, and affordable food. Furthermore, this food has to be produced in ways that do not exceed the natural resources [26]. It is essential to transform the food system to create a sustainable food system that ensures nutritious, safe, affordable, and sustainable food for everyone [27]. Incremental changes are not enough to transform the food system [3].

### 2.2. Starting Points for Food System Innovations

Caron et al. [3] categorised four interdependent parts enabling the transformation of the food system as a whole (see Figure 1).

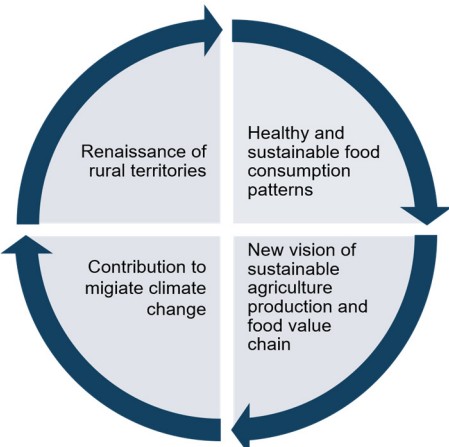

**Figure 1.** Four parts to transform food systems (own Figure, based on [3]).

First, "healthy and sustainable food consumption patterns" [3] should be established to provide nutritious and healthy diets for everyone [3]. Secondly, a "new vision of sustainable agricultural production and food value chains" needs to be created to ensure that natural ecosystems are regenerating themselves and are set for the future. All the associated services must support these visions and be tailored to the specific context and the surrounding ecosystem. Thirdly, the system ought to contribute to mitigating climate change. Lastly, a "renaissance of rural territories" emphasises the importance of a global approach [3].

Although the transformation of the food system requires many changes, a possible catalyst to alter the food system has been and will be the use of research and innovation [1]. Continuous growth in the food industry depends on innovations [28]. It is critical to develop innovations, increase the share and adaptation of innovative solutions, and develop specific solutions for a given ecosystem that can be adopted by distinct socio-economic and socio-cultural conditions [29]. These innovations can positively influence food chains or the food environment, e.g., the availability of foods [29]. However, it is also necessary to evaluate the associated risks of innovation and technologies to ensure the best use of innovation and technologies [29].

Along the entire supply chain, it is possible to make transformations according to the four parts of healthy and sustainable food consumption patterns, contributing to mitigating climate change, new visions of sustainable agriculture production and food value chains, and rural territories' renaissance.

Production is an area that is increasingly supported to find new and more sustainable solutions, for instance, agroecology, sustainable intensification or climate-smart agriculture [5]. However, each new approach comes with unique challenges. For example, the use of digital technologies is advantageous for efficiency but creates an issue of data privacy [5], or genome editing crops offers potentially higher quality crops with improved flavour profiles and lower production costs but comes with uncertain sustainability and environmental impacts [30].

In food processing and packaging, substantial advancements have been made in recent years [28]. The advances in food processing contribute to a more sustainable approach, for instance, by using less water and energy and creating higher-quality foods with better nutritional values [28]. Currently, the novel technologies in food processing are divided between thermal and non-thermal processing. Thermal processing includes microwave, ohmic heating, infrared heating, and radiofrequency heating [28]. Non-thermal food processing includes pulsed electric fields technologies, high-pressure processing, irradiation, ozone processing as well as supercritical carbon dioxide [28]. Other process technologies that are expected to be of importance in the future are Industry 4.0, barcode technologies and aseptic filling [31].

Similar to the other aspects, logistics has undergone significant changes [32]. Significant challenges in logistics follow the two main trends of extended food production seasons and improved logistics technologies [32]. Recently, the food system has been trying to adapt to the changing consumer demand of shorter food chains, increasing efficiency, better connection to producers, and having more transparency over the food they are eating [32]. However, the physical side of logistics is complemented by digital distribution systems benefitting from new technologies [32]. That there is a need for improvement was highlighted by Jorge-Vázquez et al. [33] who found a suboptimal and heterogeneous degree of digitization of European agri-food cooperatives. Similarly, Mozas-Moral et al. [34] showed that for Spain, the main retail distribution chains that operate are not solidly committed to selling organic products through their websites. Furthermore, Saetta and Caldarelli [35] demonstrated, on the basis of a case study, how the sustainability of the agri-food supply chain can be improved trough innovation in Internet 4.0. Major technologies that are important for transforming the food system are robotics and automation, big data, simulation, system integration, the Internet of Things, cybersecurity, cloud computing, 3D printing, augmented reality, ledger technologies such as blockchain and artificial intelligence [36]. According to Zhu et al. [37] the use of Internet of Things technology to realise agricultural information management can effectively ensure the safe production of agricultural products.

When it comes to innovations concerning sales and consumption, it is helpful to look at food consumption trends in Germany. Public interest in safe, natural, nutritious and sustainably produced foods has grown [38–41]. Trends with regards to food can be clustered by topic and include health, sustainability, quality, relish and everyday life aspects, as well as going beyond food, for instance with plant-based foods. A good example illustrating some of these trends is the consumption of organic foods, which has consistently increased since 2009 [38]. Consumers buy organic products due to species-appropriate animal husbandry (sustainability), getting foods that are as natural as possible (health, sustainability) and supporting local businesses [38].

However, all these innovations and trends are only starting points because first, the variety of possible solutions is so broad that it exceeds the scope of this paper and second, to tackle the crucial issues and transform the food system, novel solutions are needed that by nature, will face major resistance in the current system. Large-scale changes do not happen at the individual household level but rather by looking at industries and the resources available to the industry as a whole [42]. The transition of the food system is complex and

requires change on numerous levels [43]. Therefore, it is useful to concentrate on the food industry, its available resources, and the environment it provides for innovations to thrive.

Thus, this paper focuses on innovations that can potentially transform the food system or significantly improve the food system in one of the four areas of transformation, so-called transformative innovations [44]. It thereby concentrates on transformational innovations in contrast to other innovations in the food system, e.g., new flavour combinations.

### 2.3. Reasons of Failure for Start-Ups

In the research, the leading and most common reasons why start-ups fail fall into the same broad categories [45–47]: (1) human resource reasons (e.g., personal reasons or reasons within the team), (2) market reasons (e.g., no need for a product/service in the market), (3) financial reasons (e.g., no investments raised, wrong pricing/cost issues), (4) network reasons (e.g., expertise and know-how, assistance in obtaining funds) and (5) other reasons (i.e., lack of attention to trends, implementation risks).

Major reasons start-ups fail also depend on the stage the start-up is currently in. For instance, in the pre-seed stage, product/service development represents a considerable challenge, whereas in the start-up stage, acquiring sales and customers is key [48].

It is noteworthy that in a functioning ecosystem, all or nearly all the mentioned stakeholders, factors, and resources that are conducive to transformative innovation must be present [44]. None of the activities or actors alone are sufficient, and only through combination and interaction, transformative innovations are made possible. For instance, providing an office space without access to capital or mentoring will hinder transformative innovations.

Today, start-ups are more likely to be new market entrants compared to established companies whose main competencies are incremental innovation [12]. Many companies hold onto existing systems, products, and services for too long due to, for instance, entrenched structures that prevent creativity. Start-ups are often the originators of radical or disruptive breakthrough innovations [12,49]. It is, therefore, vital to create an environment that supports innovative and scalable start-ups [12].

## 3. Research Method and Data Sources

Expert interviews were used to determine how the German food innovation environment for transformative innovations, especially food start-ups, is structured and can be improved [50]. The study was structured as follows, which is also corresponding to the chronological sequence.

The research field was specified based on the previous literature research and the research question. A qualitative research approach was chosen as an appropriate means to answer the research question of how food start-ups are possibly different from other start-ups, how far they need a specific support environment, and how the innovation ecosystem for transformative food innovations in Germany can be improved. Similar to Garner [51], we believe that substantive issues in entrepreneurship are rarely addressed, and that many of the important questions in entrepreneurship can only be asked through qualitative methods and approaches. The technique of qualitative interviews was deliberately chosen because they make aspects of subjective experience accessible and events that are not directly observable can be recorded [50]. The live situation of the interview can create a more personal atmosphere, a better assessment of the data's quality and the background information of the interviewees can be concerned. In addition, more information can be obtained in a shorter period through the oral rather than the written expression. The expert interview and results can, thus, be understood as complementary and explicative rather than evidential or representative

Qualitative data were collected using a semi-structured interview approach. For this, (1) an interview guide was created (see Appendices A–C), (2) relevant experts were defined and contacted, and (3) interviews were conducted, recorded, and transcribed.

The written interviews were then analysed in a summarising content analysis with inductive category development using Mayring's qualitative content analysis [52] with the use of the tool MAXQDA Release 2020.4.1 (VERBI–Software. Consult. Sozialforschung. GmbH, Berlin, Germany). The analysed and categorised interviews were then presented and structured along with the research questions. Lastly, a connection between the theoretical part and the analysed empirical research was made to answer the research questions and propose recommendations.

To gain the best answers for the research questions, three types of experts were identified: (1) start-ups, (2) ecosystem network actors and (3) food system experts. The reasons why these three groups were chosen and the criteria for the selection of the experts are explained in Table 2.

**Table 2.** Overview of the interview experts and their selection criteria.

| Category | Explanation | Selection Criteria |
| --- | --- | --- |
| Start-ups | Personally involved<br>Personal experience<br>First-hand insights into challenges and support received | Location: Germany<br>Start-ups in the beginning stage (<3 years)<br>Possibly a transformative innovation<br>Range of start-ups covering the whole food value chain<br>Founding member or involved since the beginning |
| Ecosystem Network Actors | Actively shaping the start-up environment<br>Deal with a wide variety of (food) start-ups<br>Knowledge about challenges and criteria for start-up success or failure | Working at a relevant ecosystem actor<br>Ideally covering different areas in Germany<br>>2 years of experience working with (food) start-ups |
| Food System Experts | Insights into specialities when dealing with food<br>Broad overview of the general development in the industry<br>Up to date about the latest innovations and trends | Deal with innovation in the food system<br>>5 years of experience in the German food system |

An overview of the specific profiles and information of the experts is provided in Table 3.

**Table 3.** Overview of the profiles of the interview partner.

| Category | Organisation | Gender | Position | Founding Year/Time in Company/Work with Start-Ups | ID |
| --- | --- | --- | --- | --- | --- |
| Start-ups | Microplastic | Male | Founder | 2019 | ST1 |
| | Fish | Male | Founder | 2020 | ST2 |
| | Fruits | Female | Founder | Not yet incorporated, started in 2019 | ST3.1 |
| | | Male | Founder | | ST3.2 |
| | Beer | Male | Founder | 2019 | ST4 |
| | Ingredients | Female | Founder | 2019 | ST5 |
| | Plant-based dairy | Male | First Hire | 2019 (founding June, joined in October) | ST6 |
| Food system | State initiative food | Male | Management | 10 years | FS1 |
| | Research institute | Female | Management | 12 years | FS2 |
| | Research institute | Male | Management | 10 years | FS3 |
| | Research institute | Male | Director | 15 years | FS4 |

**Table 3.** *Cont.*

| Category | Organisation | Gender | Position | Founding Year/Time in Company/Work with Start-Ups | ID |
|---|---|---|---|---|---|
| Network partner | Food entrepreneur and start-up association | Male | Founder | Foundation of association in 2018, worked with start-ups before that | NET1 |
| | Accelerator agriculture, food, digitalisation | Male | Project Manager | 2 years | NET2 |
| | Research institute | Female | Project Manager | Worked with start-ups for 6 years | NET3 |
| | Food start-up incubator | Female | Project Manager | Worked with start-ups for 3 years | NET4 |

Since the focus of this study is on start-ups, most of the interviews were conducted with experts from this group. The identification (ID) is especially relevant for further analysis since all interviewees will be referred to with their anonymised ID.

The interview guide set up five topic blocks to reach general compatibility. These topic blocks are (1) background information on the person and company, (2) transformative innovation, (3) challenges for start-ups, (4) support environment, and (5) blind spots and lacks in the support environment. Start-ups were not asked about the third topic, transformative innovation, since they are assumed to be biased about their specific technology being transformative. Each topic block then had top questions as a starting point and sub-questions if the interview partner did not answer the questions holistically.

The interviews were then conducted with individuals and, in one case, with a pair, with the face-to-face interview being preferred. However, video interviews were also realised due to the still ongoing COVID-19 pandemic and the respective distance of the interview partners. An overview of the formalities of the interviews is summarised in Table 4.

**Table 4.** Overview of the interviews' formalities.

| Category | ID | Format | Language | Date | Duration (hr:min:sec) |
|---|---|---|---|---|---|
| Start-ups | ST1 | Face-to-Face | English | 7 July 2021 | 33:49 |
| | ST2 | Video Call | German | 16 July 2021 | 37:23 |
| | ST3.1/ST3.2 | Face-to-Face | English | 09 July 2021 | 30:33 |
| | ST4 | Video Call | German | 26 July 2021 | 21:13 |
| | ST5 | Video Call | German | 27 July 2021 | 39:39 |
| | ST6 | Video Call | German | 28 July 2021 | 34:41 |
| Food system | FS1 | Face-to-Face | German | 15 July 2021 | 43:21 |
| | FS2 | Face-to-Face | German | 16 July 2021 | 27:28 |
| | FS3 | Face-to-Face | German | 3 August 2021 | 41:19 |
| | FS4 | Face-to-Face | German | 9 August 2021 | 55:41 |
| Network partner | NET1 | Video Call | German | 13 July 2021 | 1:07:32 |
| | NET2 | Video Call | German | 15 July 2021 | 27:49 |
| | NET3 | Video Call | German | 3 August 2021 | 52:41 |
| | NET4 | Video Call | German | 9 August 2021 | 35:01 |

The interviews were conducted in the period from 7 July 2021 to 9 August 2021. All interviewees agreed to record the interview to facilitate further analysis.

After the interview, all recordings were transcribed based on the content semantic transcription created by Dresing and Pehl [53]. Before further analysis, the transcripts of the interviews were anonymised by replacing identifiable information with information that included the meaning and the relationship to the subject of investigation, which was marked by square brackets [54]. The type and extent of anonymisation are adapted to

the research objective of analysing the innovation ecosystem in Germany with its relevant actors, i.e., relevant actors and their location must still be identifiable [54].

*Qualitative Content Analysis*

The anonymised interview data were then analysed using the qualitative content analysis by Mayring [55]. A summarising technique with an inductive category formulation approach is applied to interpret the results and material generated in the conducted expert interviews. Summarising the content means that the content is brought down to such an extent that only the most important aspects and essential components are retained [52]. The approach follows the structure proposed by Mayring (28) and was executed using the analytical tool MAXQDA.

## 4. Results

### 4.1. General State of the System

At the beginning of the interview, the food system and network experts were asked about the general state of the system and their assessment of it. They explained that the food system is complex (FS3, FS4, NET1) and that the value chain is continuously growing closer together as agriculture and food are going hand in hand (NET1). The experts agree that the food system needs a transformation because the current way of conducting business is not able to satisfy future needs, for instance, nourishing a growing population or meeting the United Nations' Sustainable Development Goals (FS1, FS2, FS3, FS4, NET1, NET2, NET3, NET4). A systemic change is, thus, not only needed but also seen as inevitable (FS3, NET1, NET4). In terms of numbers, farmers and especially consumers are also the most significant drivers for a transformation (FS1, FS2) because "in the end, it (food products) needs to be bought" (FS1).

Innovations are "certainly" a key factor in this transformation (FS3, NET1, NET4), and start-ups are often a driving force for creating innovations (FS4, NET1, NET2, NET3, NET4). This is due to the nature of start-ups, as they are more explorative and not restricted by existing structures or perhaps their image and consumer promise compared to more mature companies (NET2, NET3). Nonetheless, start-ups only represent a fraction of the food system, and innovations are also developed by the established industry (FS3). However, it is also important to note that, especially in the food industry, the level of investment in R and D is low (FS3).

Furthermore, changing the food system cannot happen from one day to another but will probably be "in many small steps" (FS2). For instance, from a simple production perspective, it would not be possible to provide only plant-based proteins to the current population if everyone would decide tomorrow that they do not want to eat animals anymore (FS3, FS4, NET1). Trends and content fields, according to the experts, include alternative proteins or less animal-based and more plant-based products (FS1, FS4, NET1, NET3, NET4), health aspects such as personalised and functional foods (FS1, FS3, NET1), possibly new and more sustainable technologies (FS1, FS4, NET3), circularity (NET1, ST4), more digital approaches (FS2, NET1, NET3), a more decentralised way of producing foods (FS1) and food safety (FS3, NET4).

### 4.2. Challenges for Start-Ups in the Food System

The challenges are structured along the five broad sub-challenges: (1) external stakeholders and the market, (2) consumers and customers, (3) the people aspect, (4) structure and regulations and, (5) the start-up and product. Each of the sub-challenges has a varying number of specific challenges, each of which falls into one of the sub-categories. An overview of the categories' challenges with their sub-categories and particular challenges can be seen in Table 5. The number in the right column provides an overview of how often a specific aspect was mentioned during the interviews. If an aspect was mentioned multiple times during an interview, it was also marked multiple times, indicating the importance of the aspect.

**Table 5.** Overview of the categories' challenges.

| | | |
|---|---|---|
| Challenge: Market and external stakeholders | Investors | 6 |
| | Food system | 7 |
| | Market listing | 8 |
| | Right support, network | 11 |
| | Traditional industry | 11 |
| | Market, market fit, market need | 12 |
| Challenge: Consumers and customers | Informing customers, users | 7 |
| | Consumer changes | 8 |
| | Customers | 12 |
| Challenge: People aspect | Courage to fail, solving customer needs | 2 |
| | Personnel | 5 |
| | Being a founder, allocation of attention | 6 |
| | Knowledge exchange | 10 |
| Challenge: Structure and regulations | Structure of the support system | 4 |
| | Legal requirement, contracts | 4 |
| | Protecting idea | 6 |
| | Administration | 10 |
| | Food regulations | 20 |
| Challenge: Start-up and product | Sustainability | 3 |
| | Toll manufacturers | 5 |
| | Balance marketing and product development | 6 |
| | Business model, business development | 6 |
| | Distribution, logistics, packaging | 6 |
| | Scalability of the solution | 12 |
| | Product development | 14 |
| | Production | 17 |
| | Monetary aspects | 32 |

### 4.2.1. Challenge: Market and External Stakeholders

One aspect that was named challenging was investors (NET1, ST2). There is a power imbalance as start-ups are dependent on the investors, and investors often have more knowledge of the contracts and start-up valuation due to the sheer number of deals they are making (NET1, ST2). As a result, start-ups may be exploited and sometimes abused only to generate more capital for the investors (NET1). The food system itself can also be demanding (FS1, FS2, FS3, FS4, NET1, ST3) as it is a market with "extreme competition" over "low margins" (FS2) which provides products where "the quality is right, the price is right, [and] the security is right" (FS4). Another challenge for food start-ups is getting a market listing and remaining listed (FS1, FS3, FS4, NET1, ST4).

Furthermore, start-ups explained that it could be difficult to "get a network of people that support you" (ST3) and "that you need to find the right programs" (FS1). Furthermore, the traditional industry provides some challenges as they, on the one hand, are dependent on innovations for the future of their business (NET3) and, to a certain degree, innovate themselves (FS3, FS4). On the other hand, the established actors also feel threatened by new technologies, which can slow down start-ups as they prevent certain processes, try to imitate, or buy up innovative start-ups or ideas (FS1, FS4, NET3, ST1, ST2). Lastly, the biggest challenge in the sub-category market and external stakeholders is finding the right market, a market fit or a market need (FS1, FS3, FS4, NET2, NET3, ST1, ST3). This includes finding the right partners (ST1, ST3, ST6, NET3), the right investors (FS1, NET1) or the right programs (FS1, ST3).

### 4.2.2. Challenge: Consumers and Customers

Consumers buying food are more critical than those buying cosmetics or new technologies, especially when new processing techniques or ingredients are possibly involved (FS3, NET1). They need to be informed about the benefits of processing technologies or about

how new products provide benefits to their diet. For instance, start-ups working with algae need to show consumers the benefits while providing a product that is easy to prepare and not considerably different from what they know (NET1). Additionally, consumers and users need to be persuaded that they can make a difference and change something with their choices, even with a basic commodity such as food (FS1, FS2, FS3, NET4). Start-ups should understand their customers, talk to them, and, if needed, be willing to adapt their products and services accordingly (FS1, NET1, NET3, NET4, ST1, ST4, ST5).

### 4.2.3. Challenge: People Aspect

People need to have the courage to establish a company (FS3, ST2). Secondly, finding and attracting the right personnel for your start-up (ST2, ST6) and creating a diverse team with expert knowledge in key areas (NET1, NET3) can be challenging. Furthermore, being a founder (ST2) and allocating your time and attention to the right aspects is a challenge by itself (FS3, FS4, NET3, ST3, ST4). This shows, for example, when start-ups would like to attract investors and pitch their ideas but have not yet evaluated the market need and their specific target group (NET3). Lastly, knowledge divergence and the need for knowledge exchange provides a challenge (FS1, FS3, FS4, NET1, NET3, NET4, ST2, ST3). This means that often, founders have a great idea but do not know how to implement it properly. This is the case for scientists who do not know about the business side (ST2, ST3).

### 4.2.4. Challenge: Structure and Regulations

Aspects that are challenging for all start-ups are the structures of support systems which corresponds to the federal system and includes many programs that are not self-explanatory and mainly focus on the support of start-ups in general (FS1, ST1). Furthermore, legal aspects such as which contracts have impacts (ST2) and what needs to be respected from a legal perspective for founding or investing matters (FS1, ST2) can be challenging. This can also mean that even if a product is legally admitted in the EU, producing in Germany is not admitted whereas selling is allowed (FS3).

As for more food-specific challenges, it is tough to protect the ideas of food start-ups. "Recipes or new ingredient cannot be protected" (FS1). Food start-ups are thus, reliant on good contracts making it more difficult to steal or copy an idea (NET1). New technologies can be protected, but they may need special approvals that can be costly and time-consuming (see food regulations). Another general challenge for start-ups is the administrative effort in Germany (FS2, ST1, ST4, ST5, ST6). Lastly, almost every interview partner mentioned food regulations as a key challenge for start-ups (FS1, FS2, FS3, FS4, NET1, NET3, NET4, ST3, ST6).

### 4.2.5. Challenge: Start-Up and Product

One challenging aspect is being sustainable, finding sustainable solutions such as packaging and staying true to your goal of sustainability in every business aspect (NET4, ST3, ST4). Finding a toll manufacturer that (1) can produce your product, (2) at an appropriate amount, (3) at an affordable price, and is (4) not stealing your idea in the process is also demanding (NET1, NET4, ST6). Furthermore, it is not enough to create a product. Still, you also need to balance your marketing and product development activities, as both aspects are essential for the success of your later product (FS4, NET1, NET2, NET4, ST4). "Then everyone is happy that they are on sale, but they forgot about marketing" (NET1), and nobody is buying the product or knows about the brand. Equally challenging is creating and developing the business model (ST3, ST4, ST5). This can be due to the fact that your start-up needs to produce the products themselves (ST3), should grow sustainably (ST4) or iterate the business model (ST5). The challenge of distributing the product and finding the appropriate logistics and packaging is identically decisive (NET1, NET4, ST4). This aspect can be found, for example, in the compliance with cold chains (NET1, NET4), finding the suitable packaging as not every packaging can be used for every product (NET4), or the

storage of (rescued) products (ST4). An additional challenge, especially for food start-ups, is the scalability of a solution (FS1, FS4, NET1, NET4, ST1, ST4, ST6).

*4.3. Support Offering and Actors in the German Innovation Ecosystem*

As diverse as the challenges faced by start-ups are, so too are the support services provided in the German innovation ecosystem. This category is divided into four sub-categories: (1) the support selection process, (2) the support actors, (3) the support structure, and (4) the support offer (see Table 6).

**Table 6.** Overview of the categories of support.

| Support Category | Concrete Support Measures/Topics | |
| --- | --- | --- |
| Support selection process | Giving away equity | 3 |
| | Guidelines | 8 |
| | Added value | 8 |
| | Region and topic | 10 |
| Support actors | Government | 12 |
| | Established industry | 13 |
| | Accelerator program | 14 |
| | Investors, Venture capitalists (VC), business angels | 14 |
| | Support network | 15 |
| | Incubator program | 16 |
| | Universities and research institutes | 16 |
| Support structure | Specific vs. open support | 4 |
| | All actors are valuable and need to cooperate | 7 |
| | Regional vs. cross-regional vs. national | 9 |
| | Most/minor important actor | 15 |
| Support offer | Creating a local innovation ecosystem | 3 |
| | Equipment, product development, production | 5 |
| | Finding the right support | 5 |
| | Criticism: Investors | 6 |
| | Access to experts, industry | 9 |
| | Monetary support | 13 |
| | Network, collaboration, mentoring | 18 |
| | Providing mentoring and knowledge | 18 |

4.3.1. Support Selection Process

The support selection process describes the underlying conditions necessary for choosing or excluding an available support offer. An exclusion factor was when the programs took equity, especially if the start-up was still at an early stage (ST1, ST5). The decision for or against a specific type of support is a very individual choice for the start-ups. Still, guidance can include recommendations from other start-ups or their network (ST1, ST6), the reputation of a specific program (ST1) and if the start-up stage matched the support offered (ST5). It was also essential that the program added value either in monetary support, expertise or that the program could teach something new (ST1, ST5). The most crucial selection factor, according to the mentions, was the region and the topic of the support program (ST1, ST3, ST5, ST6). Although some ensured that the program's location was accessible and that it fit thematically (ST1, ST3, ST5), others applied all over the world, especially for digital offerings (ST1, ST3, ST5).

4.3.2. Support Actors

The actors who offer the services or support to start-ups independently or in programs are diverse. All actors named during the interviews were counted to evaluate if the identified actors were relevant. As all actors serve start-ups, the innovation ecosystem actor start-ups were not counted. Both specific actors, such as Edeka, and a whole group

of actors, such as corporates, were filtered. For each interview, an actor was counted only once, even if they appeared several times. To obtain a broad overview, see (Figure 2).

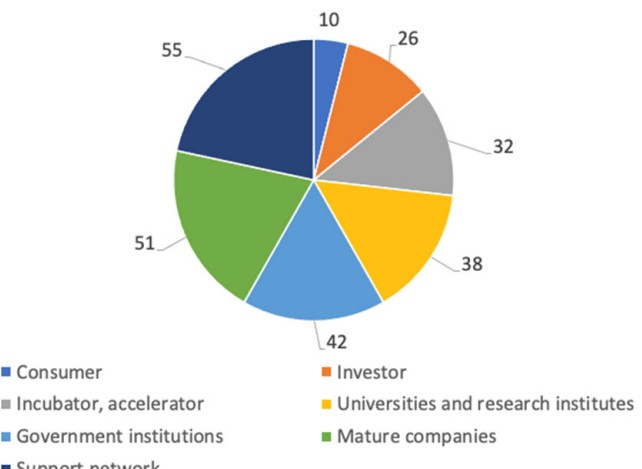

**Figure 2.** Actors mentioned by the interviewees.

Thus, a simple count of which actors were named in the interviews with the corresponding assignment to the main actors in the innovation ecosystem revealed that the support network played an important role and was mentioned 55 times with different supporters, such as experts or mentors from various fields, and start-up centres. The group of support actors is also where the most diverse actors can be classified. However, it is also easy to see that all actors can be considered relevant, and, in addition, consumers have also been identified as significant actors in the innovation ecosystem.

Although Figure 2 shows the variety and number of different actors in the system according to the six main innovation ecosystem actor groups, it also makes sense to look at the specific cooperation partners in the system to see which actor was primarily used for which support. Figure 3 provides an overview of which partners were mentioned as cooperation partners for the different interviewees.

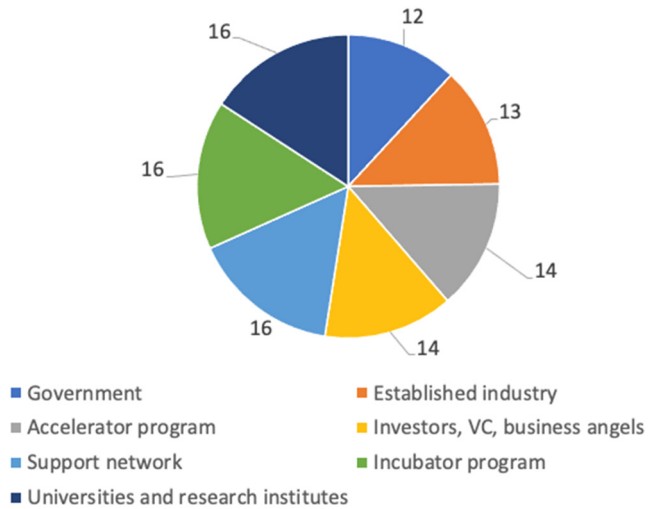

**Figure 3.** Cooperation partner mentioned by the interviewees.

Universities and research institutes were mentioned as breeding grounds for talents and ideas (FS4), for providing technical infrastructure and expertise (FS4, ST3, ST6), and as partners for cooperation (NET3, NET4, ST4, ST5). Incubator programs were mentioned as some start-ups participated in incubator programs (ST3, ST5, ST6), and interview partners offered incubator programs (NET3, NET4), especially those that provide infrastructure

knowledge about topics such as food technology, entrepreneurship or market discovery, and basic monetary support.

The support network was specified as a valuable actor for knowledge, mentoring, and a broad support network (NET1, NET3, NET4, ST1, ST3, ST6). Other critical sources, mainly for money and support and their network (ST2), are investors, VCs or business angels (FS1, FS4, NET4, ST1, ST2, ST6). Even if ST3, ST4 and ST5 are not yet cooperating and working with investors, they plan to do so in the future. Accelerator programs were also seen as an advantageous actor which, similar to incubators, provided workshops, knowledge, networks, and some monetary support (NET2, NET3, ST2, ST3, ST5, ST5).

Additionally, the established industry proved helpful and accommodating as a cooperation partner (NET2, NET3, NET4, ST1, ST2, ST3, ST4, ST5, ST6). In this context, the established industry is relevant for exchange (ST2, ST4), for knowledge sharing (ST1, ST5, ST5), as customers and suppliers (ST1, ST5, ST6), or even as strategic partners (ST4). The established industry and mature companies provide financial support and a network (NET2, NET, NET4). Lastly, the government was mentioned (FS1, FS4, NET4, ST2, ST3, ST4, ST5, ST6) for providing financial support, e.g., EXIST Business Start-up grants (ST5, ST6), a legal framework and interest group to set the issues on a political agenda (ST2, ST4).

### 4.3.3. Support Structure

According to the interviewees, the most and least important actors provided a wide range of different actors. For NET2, start-ups are the most important actors. NET3 mentioned that the most important actors are also the EIT Food consortium partners due to the nature of the structure. Although NET4 evaluated that the government and politics are the most important factors, FS4 argued that both are the least important of all the actors. For ST6 and ST2, investors were crucial, although they both suggested that it would have been possible without them but much more difficult. Some interviewees expressed that it is difficult to name the most important actor as all actors are valuable (FS2, FS4, ST3, ST5, ST6). "You need all the actors for different aim[s] or different help. And if one of them is missing or lacking, then you have (...) a problem" (ST3).

A distinct structure, i.e., regional, national and transnational offerings, and a mix of industry-specific and industry-open support, is also vital for a diversified offering. For example, industry-open support provides help on the hard facts for start-ups, such as start-up founding or funding application processes, and industry-specific assistance is helpful for the specific issues and characteristics of an industry (NET3).

### 4.3.4. Support Offer

The support offered by the actors covers a wide range of different aspects. With their offering, the network actors try to benefit start-ups and create and improve the local innovation ecosystem of their respective areas (NET2, NET3). Equipment and test kitchens for small scale productions are provided (NET4, ST6), and the support actors help start-ups find the proper assistance they currently need (NET1, NET2, NET3, NET4). To do this, they refer start-ups to other actors who have the required support (NET3), open doors for the start-ups in their network (NET1, NET2), and help with project applications (NET4). However, the support provided by investors was also criticised by NET1, ST2 and ST4 as investors can exploit start-ups and their unawareness concerning some aspects. Thus, investors and the conditions of their contracts need to be scrutinised. Additional support provided by the actors is access to experts and the industry, either by the support actors themselves or through experts in their network, coaches, and workshops (NET2, NET3, NET4, ST5, ST6). To a certain extent, monetary support is also provided during programs and by actors in the innovation ecosystem (FS1, NET2, NET3, NET4, ST2, ST4, ST5). However, although getting capital is not as easy in Germany as it may be in other countries, there is still a lot of monetary support available (FS1). Crucial support given to the start-ups is the network, fostering collaborations with and within the network and mentoring start-ups (NET1, NET2, NET3, NET4, ST1, ST3, ST4, ST5).

Generally, the interviewees agreed that the support already available in Germany is pretty good, and the possibilities to get help are numerous (FS1, FS3, FS4, NET1, NET2, NET3). This was also substantiated by the start-ups, which agreed that they received valuable support and could get into helpful programs (ST1, ST2, ST6). Beyond that, the support system is also developing and is "definitely driving in the right direction" (NET2) with their development (FS2, FS3, NET1, NET2, NET3, NET4, ST1, ST3, ST5). However, the evaluation also shows that there is still room for improvement and that, generally speaking, there are not as many food-specific support programs (ST6).

*4.4. Success Factors for Start-Ups and Support Actors*

According to the experts, success factors provide insights into the support and resources the interviewed start-ups used or needed. Thus, success factors give valuable findings to evaluate the innovation ecosystem and find possible improvements. The success factors are categorised into (1) people aspects, (2) hard, and (3) soft success factors (see Table 7).

**Table 7.** Overview of the categories of success factors.

| | | |
|---|---|---|
| People success factors | Focus and long-term commitment | 5 |
| | Personal experience, knowledge, skills | 8 |
| | Personal interest in mission and success | 8 |
| | Willingness to learn and inform yourself | 8 |
| | Access to experts | 10 |
| | Team | 12 |
| | Network, personal and industry connections | 26 |
| | Cooperation and exchange | 38 |
| Hard success factors | Access to tools, technical infrastructure | 5 |
| | Infrastructure and location | 7 |
| | Monetary support, financing | 9 |
| Soft success factors | First customers | 3 |
| | Transparency and honesty | 3 |
| | Diverse involvement | 4 |
| | Creating awareness and traction | 5 |
| | Luck and chance | 8 |
| | Available support | 14 |

4.4.1. People Success Factors

When it comes to the support provided by people, it is vital that they are interested in the long-term success of the start-up (FS3, NET3, ST5). This also means that the start-up should be thinking long-term and focusing on its efforts (ST1, ST2). Furthermore, it proved to be much more valuable if the people working with and supporting a start-up are personally involved and want the start-up to succeed because they believe in the idea and the mission behind it (NET2, ST2, ST4, ST5, ST6). The knowledge and skills of the founder, the founding team, or generally the start-up team are also crucial (ST2, ST4, ST6). For instance, personal knowledge about specific food retailing or business processes is much more valuable than getting the support third hand (ST2, ST4). Furthermore, if certain knowledge or skills are not currently present in the team, the willingness to learn, research and acquire certain skills or even personnel can be essential for the start-up's success. This also means that you have access to people and experts (NET3, ST1, ST2, ST3, ST6) to "get (...) the ground reality" (ST1) from people that know what they are talking about. Another success factor is the team itself, with its composition and working atmosphere (NET1, NET2, ST2, ST3, ST4, ST5). Similar to a successful IT team that needs an IT expert, a thriving food start-up probably needs a food technologist (NET1). The network, including personal and industry connections, plays a role as well (FS1, NET1, NET2, ST1, ST2, ST3, ST4, ST5, ST6). There is plenty of help available, and your network can work "like a snowball, the more you know, the more suggestions you are getting" (ST3). Lastly, all

interviewees agreed that cooperation and exchange are the key success factors for start-ups (FS1, FS2, FS3, FS4, NET1, NET2, NET3, NET4, ST1, ST2, ST3, ST4, ST5, ST6).

### 4.4.2. Hard Success Factors

Food start-ups' first hard success factors are access to tools and technical infrastructure to test out ideas and develop the product (FS1, NET1, ST4, ST6). Going in the same direction, access infrastructure and the location of the start-up are also decisive factors for success (FS2, NET4, ST2, ST5). Infrastructure refers to aspects such as the availability of living and working space or fast Internet (FS2), but also access to production facilities in the immediate vicinity (NET4, ST2), or the possibility of working in coworking spaces in general (NET4, ST5). Lastly, monetary support and financing possibilities are key success factors (NET2, NET4, ST1, ST2, ST4, ST5, ST6) because without money, "you cannot execute" (ST1). This monetary support can be provided in the form of grants, prizes, money, or by favouring payment terms.

### 4.4.3. Soft Success Factors

The soft success factors include getting your first customer that is using the technology which, if a product is truly disruptive, can spread as a "sure-fire success" (FS1). "Honesty and transparency", for instance, towards partners regarding when the product will be ready (ST2), between cooperation partners about what can be achieved (NET3), or towards your customers (ST4), is also a key for success. A diverse involvement, meaning getting "a lot of support from different sides and directions" (ST1), can also be a driver for success (FS3, NET4, ST1, ST6). Furthermore, creating awareness and traction for your start-up is vital for the success of your start-up (FS1, ST1, ST2, ST4). This can happen, for example, by bringing the problem you are solving to the attention of politicians (ST2, ST4), but also when using social media trends (FS4) or getting attention by winning a contest (ST1). Another factor that cannot be influenced as easily but that was also frequently mentioned is luck and chance (FS1, ST2, ST3, ST5, ST6). Often, the start-ups feel lucky that certain circumstances have led them to a great business partner or coach (ST2, ST3, ST5, ST6), and once a company is established and running, many things fall into place on their own (FS1). Lastly, the available support was also mentioned as a success factor (NET2, NET3, ST1, ST2, ST4, ST5). In particular, the EXIST scholarship was highlighted as valuable for the initial stage (ST5, ST6). The availability and participation of programs were deemed to be vital for success (NET2, NET3, ST1, ST2), and even if the start-up did not participate in programs, they still advocated for the available support (ST2).

### 4.5. Improvements for the Support of Food Start-Ups

The possible improvements were filtered out of the interviews as answers to what may be missing, frustrating, or improved in the current system. The category improvements are divided into four sub-categories: (1) infrastructure and support offer, (2) structural improvements, (3) cooperation improvements and (4) cultural improvements. As for the previous categories, it is also a mixture of general improvements that could help all start-ups and could also be specific improvements for food start-ups. The following Table 8 presents an overview of the improvement categories.

**Table 8.** Overview of the improvement categories.

|  | Access to the right experts, network | 5 |
| Infrastructure and support offer | Market | 6 |
|  | Expanding support | 7 |
|  | Technical support | 8 |
|  | Monetary support, financing decision | 19 |

**Table 8.** *Cont.*

| | | |
|---|---|---|
| Structural improvements | What you learn in school | 4 |
| | Using and improving established structures | 7 |
| | More manageable and more official network | 8 |
| | Less bureaucracy, administration | 9 |
| | Revising regulations | 9 |
| Cooperation improvements | Better connected ecosystem of actors | 4 |
| | Cooperation between actors | 8 |
| | More transparency | 8 |
| | Learnings from other countries, thinking across borders | 18 |
| Cultural improvements | Stronger support of female entrepreneurs | 2 |
| | Establishing founding mentality from a young age | 6 |
| | Innovation culture | 19 |

### 4.5.1. Infrastructure and Support Offer

Improvements in the infrastructure and support offer include access to the right experts and network (FS2, FS4, NET3). This refers to getting in touch with legal experts, especially in food law (FS2), and for start-ups to already think about industrial processes at an early stage (FS4, NET3). Improvements could also be made at the market level by facilitating market access, for instance, by providing a real-world retail setting to test your product (ST6), a program that would ease the market entry and acceptance of products (NET1, ST1). Another improvement could be to improve the market itself by considering fairer prices and digitalising the German market (NET1). Furthermore, some support offers could also be expanded, focusing on a long-term impact rather than being opportunity-driven (NET3). Additionally, more programs should focus on aspects other than business, i.e., production, and grants or other government programs should be offered in all states (ST3, ST5). In particular, the range of technical support, i.e., production facilities and test facilities, should be further expanded, and more technical experts should be advised early in the start-up process (FS4, NET4, ST1). Even though the Food Startup Incubator provides this support, NET4 would like "more opportunities to be able to produce your product more cheaply, so that (...) you have several production locations where you can simply try something out". Lastly, accounting for almost the most mentions is improving the monetary support (FS1, FS3, FS4, NET1, NET2, ST1, ST4, ST5). NET3 finds clear words here, because "more money is needed in the market, specifically significantly more venture capital. There is still far too little risk capital compared to abroad." However, this money should be carefully invested, leaning on existing structures and not simply "thrown into the market" (NET2), especially if it is taxpayers' money (NET2).

### 4.5.2. Structural Improvements

The second sub-category involves structural improvements. First, some interviewees would like to educate children earlier and improve what they learn in school, focusing on problem-solving skills, teaching the structure of the food system, including possible impacts on the system, and developing entrepreneurial skills in school (FS2, FS3, NET2). Second, FS2, FS4 and NET2 explain that improvements to the system can also be made by better using and connecting already established structures. Additions to the support system should thus be made by "always looking at what existing structures are already there" (NET2). Networking is another specificity that could be refined (NET3, NET4, ST5, ST6). This could be in the form of, for instance, a program that brings together directors of established businesses with founders (NET3), but also by providing a more official and institutionalised way to get in contact with other start-ups and find partners (ST5, ST6), for example, finding consultants in a more official and structured way than by luck (ST5) and sharing start-up experiences on a peer-to-peer level (ST6). Reducing bureaucracy and administration would serve all types of start-ups (FS1, FS2, FS3, NET1, NET4, ST4). This includes founding processes that "take far too long (...) especially if [you] want to

get started right away" (NET2) but also processes such as writing application forms (FS2, NET4) which is currently very lengthy and in parts "made up out of thin air" "NET4). Equally mentioned was the wish for revising regulations (NET1, ST3, ST5, ST6). Although some food regulation processes and stricter consumer protection laws were especially criticised, the aspect of employee stock options in particular was seen as an issue that could have a big impact on attracting talents (ST5, ST6). "In Germany, this [employer stock options] is taxed as income, 50%, while the founders somehow only pay capital gains tax, 25% of something like that, on what they take with them. In Great Britain, for example (...) this is equalised. (...) it makes it more difficult for start-ups to incentivise their employees because, with the low salaries that exist in start-ups, it plays a big role in terms of potential profits" as ST6 explains it.

Another challenging aspect for potentially transformative innovations is the 19% value-added tax on vegan alternatives such as plant-based milk compared to 7% for cow milk and other products considered basic needs (ST6).

### 4.5.3. Cooperation Improvements

The following sub-category deals with improving the cooperation between the actors. Although the specifications go in a very similar direction, they all deal with slightly different aspects. First, the ecosystem actors should be better connected (FS2, NET2, NET3, ST1) as the "ecosystem (is) developing food but still not everything is connected" (ST1). For instance, this applies to investors (FS2) and transdisciplinary connections, e.g., at the university level (NET2). Furthermore, more cooperation could be between the various ecosystem actors (FS1, FS2, NET3). Often the actors in the innovation ecosystem compete with each other, which partially leads to "demarcation and poaching "of start-ups" (FS1) and a situation in which the various actors also like "to steal a bit of the show from each other" (NET3). Thus, the actor in the innovation ecosystem should focus on providing the best support for start-ups and cooperation. This collaboration should also be honest and transparent (NET3), which would improve the working atmosphere and provide more transparency to the start-ups, which then know where to go and what to expect.

Additionally, the general overview of the available support offerings could be refined by making it more transparent (FS3, NET1, NET3, ST2, ST5). There are a lot of offers, possible funding opportunities, different contract forms or company forms. Still, often the knowledge and the overview might be missing (FS3, NET1, NET3), and for example, "the general funding database that exists is confusing" (ST5). Lastly, it is also possible to learn from other countries and copy the things working well in other countries (FS2, FS3, FS4, NET1, NET4, ST2, ST4, ST5). Start-ups should be inspired by the ideas and products provided in other countries and copy them (FS3) just as ST2, who got inspired by another start-up from the United Kingdom. As FS3 explains, we should think across borders as "there are innovations in other countries that are not so bad. There are smart people everywhere so that you can work with their ideas" (FS3). Countries also provide learnings and examples. For instance, the United States of America or Israel (NET4, ST2, ST4, ST5) were mentioned as an inspiration for a successful start-up innovation culture, including the availability of venture capital. Still, Austria (NET1), especially the region Vorarlberg, provides an example of the start-up scene, the support offer, and economic growth.

### 4.5.4. Cultural Improvements

The last sub-category focuses on cultural improvements. Two women mentioned that they would like to see more support for female entrepreneurs (NET3, NET4), whereas none of the male interviewees found this aspect noteworthy. Another cultural aspect that could be improved is creating an innovation culture (FS1, FS2, FS4, NET1, NET3, NET4, ST2) and a founding mentality already from a young age (FS1, FS2, NET2, NET4, ST1). Failure should not be seen as something negative, and more people should "have the courage to fail" (NET4) and start doing. This applies to founders (FS1, FS2, ST2), but also to established companies, which should be open to innovations, and investors, putting their money into

visionary ideas (FS1, FS4), being less risk averse (NET1) and freeing themselves from the fact that failed founders should not be supported again (FS1).

The corresponding categories summarise the challenges, available support opportunities provided by the different actors, success factors to see where the support system is already well established and find possible blind spots for improvements of the system. As a basis, the scene is set with background knowledge, information about transformational innovations, and general remarks. Thus, the interviewees' statements were filtered and sorted to ultimately obtain an overview of the innovation ecosystem and to identify potential improvements.

## 5. Discussion and Conclusions

### 5.1. Challenges of Food Start-Ups in the Food System

Recent research [45–47] clustered the main challenges around the topics of (1) human resource reasons, (2) market reasons, (3) financial reasons, (4) network reasons and (5) other reasons. When comparing these challenges with the main issues mentioned during the interviews with (1) market and external stakeholders, (2) consumers and customers, (3) people aspects, (4) structure and regulations, and (5) start-up and product, there are some overlaps but also some differences. It becomes apparent that, especially for food start-ups, structure and regulations and issues concerning the start-up and product are incredibly challenging.

A recent study [48] suggests that currently, the main challenges lay in sales/customer acquisition, product development and capital procurement. The reasons why start-ups fail go in a similar direction: having no market need, running out of money, not having the right team, and getting outcompeted. These challenges are also mentioned in the empirical research of this study. However, not as the most challenging aspects. Monetary aspects, food regulations and production, and product development-related issues are significant challenges for food start-ups. This is partly because food start-ups, more often than IT start-ups, work with physical products, and are rather comparable to hardware products. Primarily if a start-up deals with possibly revolutionary and transformative aspects, the chances are high that they need highly technical machinery which requires strategy-fulfilling financing rather than strategy-determining start-up financing. For instance, it can mean that one machine such as a bioreactor costs a minimum of EUR 100,000 (ST2). Additionally, the food sector is a demanding industry to enter as regulations, i.e., safety, hygiene, or labelling, are high. The products available on the market have high quality while being comparably cheap, which means low margins on the producer or seller side.

Nevertheless, it requires the use of expensive machinery, skilled labour, and experts in the field of food, making it challenging to produce a product. Furthermore, if a product or minimum viable product is successfully developed, it does not necessarily mean that it can be produced on a large scale

### 5.2. Support Offering and Actors in the German Innovation Ecosystem

Recent research found that the innovation ecosystem in Germany already provides numerous support opportunities and many actors are involved in the process [44].

Similarly, the findings demonstrate that the cooperation and interaction of the different actors are crucial. All actors identified have an impact on both the success of innovations and the change of the food system. However, as there were certain tendencies of one actor being more important, the different actors also had inclinations about which actors were the most important. For instance, for a program financed by a university, the university is the most important actor, or for an offer supported by members, those members are the most important actors, irrespective of whether these members are mature companies, universities, or research institutes.

All actors are considered crucial for success as each actor has unique benefits that the start-ups need to flourish. These preferences are also highly dependent on the structure and financing of the support offer itself. For example, although investors can offer the

necessary financing, especially long-term financing, universities, and research institutes can be considered a breeding ground for ideas and provide some basic instruments for realising the concept, and incubators and accelerators help shape and strengthen the concept by providing mentoring, basic financial support, or other resources.

Against this background, it is not surprising that network, cooperation, and mentoring are the key success factors and the primary support provided. Thus, the actors set the success of the start-ups as their top priority and try to support them with their expertise or, if necessary and possible, with the expertise of partners from their network or connections to other networks. It also does not matter whether this kind of support comes from a network that the founder or start-up team has already built themselves, e.g., through their own work experience, or whether it is made possible by programs and other providers. However, it is important that the actors focus on the needs of the start-ups, ideally in a flexible manner, and that they are interested in the long-term success, which does not necessarily entail a financial component, but also because they believe in the idea or the mission of the start-up.

Additionally, the current innovation ecosystem offers monetary support from various sides. First, minor financial funding is provided by accelerator or incubator programs or official calls for projects. Second, prizes provide some monetary benefits. Thirdly, cooperation partners such as mature companies or research institutes can offer financial support through grants or favourable financing terms. Lastly, investors, venture capitalists, or business angels offer financing support. Although investors can usually offer more financial resources and thereby potentially a faster growth of the start-up, the underlying motives of investors are not known. This, therefore, requires the support of affordable juridical help or knowledge about the potential upsides and pitfalls of investor-based financing. In the end, the right financing decision, e.g., either strategy-determining financing or strategy-fulfilling financing, depends on the start-up's specific circumstances and goals. There are no right or wrong choices as all options can be valuable.

Generally, the support available in Germany is perceived as excellent and extensive. This includes an industry open offer and industry-specific food offerings. Both offerings provide value to start-ups and promote the potential success of food start-ups. Most of the offerings are open and thus focus on the general foundation of a successful start-up or the business side. This is especially useful for all the first-time founders with no background in business, such as scientists or food specialists. However, the range of food-specific offerings is expandable, as, during the interviews, the same actors were repeatedly mentioned.

Additionally, this food-specific offering is crucial for founders who do not have a food background. Therefore, the knowledge and expertise on the market are plentiful. It is to a certain extent also dependent on the founders themselves, whether they acquire this knowledge, seek the exchange with experts, but are also put in contact with the right experts. Nevertheless, establishing these connections and trying to support start-ups as best as possible is part of the guidance offered by existing innovation ecosystem actors.

The granted support is, therefore, as diverse as the start-ups, the products or services they contribute, and the needed assistance. It also does not matter which of the different actors offer the support if someone in the innovation ecosystem provides it. A mixture of regional and transregional offers is also valuable, as regional offers provide direct feedback but cannot cover every needed expertise.

### 5.3. Improvements for the Support of Food Start-Ups

As the literature already emphasises [12,49] a successful innovation ecosystem is characterised by a well-organised, collaborative, and supportive environment with a vivid exchange between the members in the ecosystem. The interviewees confirmed this, and although the different actors are already cooperating, there is still room for improvement.

The most common recommendation for improving cooperation is learning from other countries such as the United States of America and Israel and bringing the best to Germany. This includes suggestions for politicians, for instance, by looking at the Austrian region

Vorarlberg, and suggestions for investors, e.g., by providing more capital, and for start-ups who should bring innovations from other countries to Germany. The support available is extensive, especially for general industry support. Therefore, a more practical approach to improving the whole system is increasing the cooperation between the actors. This could be implemented by bridging the gap between industry-specific support, i.e., for food start-ups, and open industry support and cooperating to create even better support offers. Additionally, all the actors in the innovation ecosystem should collaborate more and not see other actors as competitors but rather as partners, providing the best support for start-ups. For instance, working transparently, sharing your competencies, and being honest about what you can or cannot offer.

The second improvement to the system concerns structural improvements. Especially when it comes to decreasing bureaucracy and administration, these changes are difficult to implement in the short term. However, an explicit recommendation of this paper is to revise tax regulations, especially concerning employee stock options, to reduce bureaucracy, e.g., founding processes, and to create structures that represent reality. Food regulations should also be reconsidered, but this is even more of a challenge because, on the one hand, regulations apply equally to small and big companies and exemptions for small companies would create an unfair advantage, and, on the other hand, they also ensure the food quality and safety that we currently have. Thus, finding solutions for these changes probably requires more long-term approaches. However, immediate strategies to improve the system can be made by providing support concerning these issues, e.g., low-threshold and low-cost access to relevant experts. This could also imply creating more institutionalised or official networking options or platforms where like-minded people can meet, start-ups can exchange experiences, and experts or consultants can be found or offer their help. This would also eliminate the success factor of luck to a certain extent, since start-ups would no longer have to rely on others to find possible coaches and experts but would find the support they need themselves. Additionally, a more interwoven and connected support system would help in this aspect.

Lastly, a more complex aspect of change is creating a culture that favours and encourages innovation. Although culture is difficult to change, it is also a key aspect that the interviewees wished for. The actors wish more people would just go for it and think less about possible consequences and failures. In school, children should be taught about the basics of running a business, and teaching should be focused on problem-solving skills and independent thinking. In general, failure should not be seen as something bad but rather as an opportunity to learn. Fortunately, there is already a positive development in an innovation culture in Germany. There is ample support in the system that wants innovative ideas to succeed. Nevertheless, although some aspects are more difficult to change than others, an ecosystem is by design constantly evolving and adapting to new opportunities and circumstances, which means that the potential for changes in the system is high.

## 6. Direction of Future Research

Future research could focus on the background of the founder and further explore what types of founders exist, both in food start-ups and in other types of start-ups. This could help to explore how the background, and the team constellation have an impact on the necessary support or success of a start-up. In addition, further exploration of other subcategories of start-ups also seems useful to identify any divergences and synergy opportunities. The influence of the pandemic and the extent to which offerings were shifted to digital, were adopted by start-ups, and thus may have further reduced the location factor in support offerings, which would also be an interesting possibility for future research. From another perspective, it is certainly insightful to interview start-ups that were not successful and to find out what the reasons for the failure were and what might have helped or reduced such failures. Additionally, a larger scale quantitative study where explicitly food start-ups or other industry-specific start-ups are surveyed about their challenges,

success factors and the evaluation of the support offered would generate new insights for the start-up scene and the innovation ecosystem in Germany.

## 7. Limitations

The rare available literature and lack of information about food start-ups was a limitation in this study. Most academic literature focuses on start-ups in general and if industry-specific papers exist, they concentrate on industries such as finance or IT. Based on this issue, the definition of food start-ups as chosen in this paper is also a point of criticism, because whether agricultural start-ups or biotechnology start-ups can ultimately be understood as food start-ups is questionable.

The focus on transformative innovations can also be criticised, as this is a very subjective assessment, and disruptive and radical innovation is difficult to identify. Whether the selected start-ups will ultimately transform the system will be seen in the future, but certainly they will have a positive impact on the system. However, the variety of start-ups and innovations that potentially transform the system is much greater than originally expected and perhaps presented in the literature.

The intersubjectivity and testability of the qualitative content analysis must again be seen as a possible limitation. Although it was strongly rule-guided, it becomes apparent that during the interpretation, there is a certain degree of leeway and interpretation of the author of this study. Certain statements that have been sorted into one category could be categorised differently in a third-party verification or analysis. For example, frustrations with the system may be seen as a potential for improvement but also as a challenge.

Lastly, the analysis only surveyed start-ups that have been successful to date, which can potentially lead to a failure to assess the true problems and reasons why start-ups fail and the associated potential for improvements in the system. Furthermore, even though the start-ups were selected to be founded at approximately the same time, they were nevertheless at very different stages with their start-ups. This can also be a point of criticism of the analysis, as the comparability can be questioned. However, this may also have subconsciously opened the space for an even greater variety of problems.

**Author Contributions:** Conceptualization, C.H., A.M. and V.H.; methodology, K.L.; writing—original draft preparation, K.L., A.P. and M.H.S.; writing—review and editing, K.L., A.P., M.H.S. and C.H.; supervision, A.M., C.H. and M.H.S.; project administration, A.M. All authors have read and agreed to the published version of the manuscript.

**Funding:** The article processing charge was funded by the Baden-Württemberg Ministry of Science, Research and Culture in the funding programme Open Access Publishing.

**Institutional Review Board Statement:** Not applicable.

**Informed Consent Statement:** Not applicable.

**Data Availability Statement:** Data are available from the corresponding author on request.

**Conflicts of Interest:** The authors declare no conflict of interest.

## Appendix A. Interview Guide for Start-Ups

For the start-ups, identical interview guides in English and German were created. Each interview guide also has a comprehensive version with all sub-questions and a shorter version that was send to the interviewees in advance.

*Appendix A.1. Interview Guide for Start-Ups (English)*

**Table A1.** Interview Guide for Start-Ups (English).

| | |
|---|---|
| **Introduction** | • Personal introduction<br>   ○ Thanking them for participating<br>   ○ Introduction of my person<br>• Introduction of the topic<br>   ○ Goal of the master thesis<br>   ○ Focus of the master thesis<br>   ○ Definitions<br>• Organization<br>   ○ Again: asking permission for recording the interview<br>   ○ Assuring confidentiality/ anonymity<br>   ○ Offering them the results of my findings |
| | Starting the recording |
| **(Narrative part): Background information on the person and company** | • Personal background: Tell me something about you as a person, your motivation and background:<br>   ○ What is your education?<br>   ○ What was your motivation for founding the start-up?<br>• Founding history: Continue with your start-up, its history and development?<br>   ○ When did you found the start-up?<br>   ○ How many people were in the founding team?<br>   ○ How many people are currently employed?<br>• Food Start-up: Do you identify yourself as a food start-up? If yes, where along the food value chain would you place your company?<br>• "Transformational innovation": which problem do you want to solve, and do you think this has a significant impact on the food system as a whole?<br>   ○ What is the problem you want to solve?<br>   ○ How do you solve this problem?<br>   ○ To what degree do you think you are innovative? (e.g., product, process, business model)<br>   ○ To what degree do you think you improve the current food system?<br>   ○ What impact do you think your innovation has? |
| **Challenges for start-ups** | • Challenges: Which challenges did you face during the founding?<br>   ○ What were the biggest challenges you faced during the time as a founder?<br>   ○ Were there specific challenges for you because you were a food start-up? If yes, please explain how they looked.<br>      → Example: food law, quality management patents, marketing, consumer research, capital<br>• Stages: Were there different challenges at different times of the process?<br>   ○ Which challenges did you have during the different stages: ideation, pre-seed, seed, start-up?<br>   ○ Which resources were helpful at the different stages of the business?<br>      → Example: physical, financial, human, intangible (patents, trademarks, knowledge, data) |

**Table A1.** *Cont.*

| | |
|---|---|
| Support environment | • Support: Have you been supported during?<br>　○ Have you been supported in the process? If yes, how?<br>　○ Did you use a specific program? If yes, which?<br>　○ What other support offers did you take advantage of?<br>　○ Were you satisfied with the offers? If yes, why/If no, why not?<br>　○ Do you think that the offer could have been improved? If yes, how?<br>• Access:<br>　○ How did you become aware of these services and programs?<br>　○ Did you look at the geographic constraints of a program? E.g., regional programs, Germany-wide programs<br>　○ Did you look specifically for food-specific programs?<br>• Exclusion:<br>　○ Did you know of a support offer, but deliberately chose not to use it?<br>　○ Did you reject support that was offered to you? If yes, why?<br>• Actor Cooperation<br>　○ During the process, did you actively reach out for one of the following actors: universities/research institutes, established companies, government institutions, incubators/accelerators, investors/VC/business angels or other support networks? If yes, which one? Why and what for? |
| Blind spots and lacks in the support environment | • Blind spots:<br>　○ Were there situations in which you would have liked support but did not receive support?<br>　○ Were there blind spots specifically for you as a food start-up?<br>　○ On a scale from 1–10 (1 is the lowest score and 10 is the highest score), how would you rate the support offers for:<br>　　■ Start-ups in Germany<br>　　■ Food start-ups Germany<br>　○ Do you know of a support offering from other places that you wished existed in your region?<br>• If you could make a wish or dream big: what would you do to shape, improve, or change the support for food start-ups in Germany?<br>• In retrospect:<br>　○ Was there support you wished you had? Anything you missed?<br>　○ Was there a program that you now know which you wished you knew about sooner? If yes, why, and how could the offer be better made available?<br>• Status:<br>　○ In which stages are you currently in?<br>　○ What support do you currently need? |
| Closing | • Anything you would like to add or forgot to mention before?<br>• Thank you for your time. Is there anything you would like to know from me? |
| | Stopping recording |
| Farewell | • Further procedure (e.g., reminder to sign the declaration of consent, when to expect results, confirming contact details, etc.)<br>• Thanking them again<br>• Saying goodbye |

*Appendix A.2. Interview Guide: Start-Ups (English, Short)*

Background information on the person and company

- Personal background: Tell me something about you as a person, your motivation and background (education, etc.)
- Founding history: Continue with your start-up, its history and development?
- Food Start-up: Do you identify yourself as a food start-up? If yes, where along the food value chain would you place your company?

- "Transformational innovation": which problem do you want to solve, and do you think this has a significant impact on the food system as a whole?
- What impact do you think your innovation has?

Challenges for start-ups

- Challenges: Which challenges did you face during the founding?
- Stages: Were there different challenges at different times of the process, e.g., ideation stage and prototyping stage? Which?
- Which resources were helpful at the different stages of the business? (Patents, trademarks, knowledge, data)

Support Environment

- Support: Have you been supported during the whole process?
- Access: How did you become aware of these services and programs?
- Exclusion: Did you deliberately chose not to use certain programs? Why?
- During the process, did you actively reach out for one of the following actors: universities/ research institutes, established companies, government institutions, incubators /accelerators, investors/VC/business angels or other support networks? Which ones? Why and what for?

Blind spots and lacks in the support environment

- Blind spots: Did you feel well supporting during the whole process?
- If you could make a wish or dream big, what would you do to shape, improve, or change the support for food start-ups in Germany?
- In retrospect, was there support you wished you had? Anything you missed? Was there a program that you now know which you wished you knew about sooner? If yes, why, and how could the offer be better made available?
- Status: Which stages are you currently in with your start-up?

*Appendix A.3. Interview Guide: Start-Ups (German)*

**Table A2.** Interview Guide: Start-Ups (German).

| Einleitung | <ul><li>Persönliche Vorstellung<ul><li>○ Für Teilnahme bedanken</li><li>○ Vorstellung</li></ul></li><li>Vorstellung des Themas<ul><li>○ Ziel der Masterarbeit</li><li>○ Fokus der Masterarbeit</li><li>○ Definitionen</li></ul></li><li>Organisatorisches<ul><li>○ Wiederholen: Erlaubnis zur Aufzeichnung einholen</li><li>○ Anonymität erklären</li><li>○ Ergebnisse anbieten</li></ul></li></ul> |
| --- | --- |

**Table A2.** *Cont.*

| Aufnahme starten |
| --- |

**Themenblock 1 (Narrativer Teil): Hintergrundinformationen zur Person und zum Unternehmen**

- Persönlicher Hintergrund: Erzähl mir doch etwas über dich als Person, deine Motivation für die Startup Gründung und deinen Hintergrund (Ausbildung, etc.)
  - ○ Was ist deine Ausbildung?
  - ○ Was war deine Motivation für die Gründung des Startups?
- Gründungsgeschichte: als nächstes erzähl mir doch etwas über das Startup, die Entstehungsgeschichte und die Entwicklung?
  - ○ Wann wurde das Startup gegründet?
  - ○ Hast du das Startup alleine gegründet? Wie viele Personen waren im Gründungsteam? Warst du Teil des Gründungsteams?
  - ○ Wie viele Personen sind derzeit angestellt?
- Food Startup: Identifizierst du dich (ihr euch) als Food Startup? Wenn ja, wo entlang der Lieferkette platziert ihr euch?
- "Transformative Innovationen": Welches Problem wollt ihr mit dem Startup lösen und denkst du, dass das einen signifikanten Einfluss auf das gesamte Lebensmittelsystem hat?
  - ○ Was ist das Problem, dass durch das Startup gelöst wird?
  - ○ Wie wird das Problem gelöst?
  - ○ Inwieweit denkst du, dass dein Startup innovativ ist? (z.B. mit Bezug auf Produkte, Prozesse, Geschäftsmodelle)
  - ○ Inwieweit denkst du, dass dadurch das Lebensmittelsystem verbessert wird?
  - ○ Was glaubst du, welche Auswirkungen diese Innovation haben wird?

**Themenblock 3: Herausforderungen für Startups**

- Herausforderung: Welchen Herausforderungen standest du während der Gründung gegenüber?
  - ○ Was waren die größten Herausforderungen, denen du während der Zeit als Gründer*in gegenüberstandest?
  - ○ Gab es deiner Meinung nach spezifische Herausforderungen, weil ihr ein Food-Startup seid? Wenn ja, welche?
    - → Beispiele: Lebensmittelgesetze, Qualitätsmanagement, Patente, Marketing, Market Research, Kapital, etc.
- Phasen: Gab es unterschiedliche Herausforderung zu verschiedenen Zeitpunkten im Prozess, also z.B. in der Ideenphase und die Prototypen-Entwicklungsphase?
  - ○ Welche Herausforderungen hattest du an verschiedenen Zeitpunkten, z.B. Ideenphase, pre-seed, seed, startup
  - ○ Welche Ressourcen waren hilfreich während der verschiedenen Phasen des Startups?
    - → Beispiel: physisch, finanziell, Personal, immateriell (Patente, Trademark, Wissen, Daten)

**Themenblock 4: Förderungsumgebung**

- Unterstützung/ Förderung: Wurdest du während des gesamten Prozesses in irgendeiner Weise unterstützt?
  - ○ Wurdest du während des Prozesses unterstützt? Wenn ja, wie?
  - ○ Hast du an einem speziellen Programm teilgenommen? Wenn ja, welches?
  - ○ Welche anderen Förderungs- oder Unterstützungsangebote hast du in Anspruch genommen?
  - ○ Warst du mit dem Angebot zufrieden? Wenn ja, warum? Wenn nein, warum nicht?
  - ○ Denkst du, dass das Angebot hätte verbessert werden können bzw. dass man das Angebot verbessern kann? Wenn ja, wie?
- Zugang: Wie bist du/ihr auf die Angebote und Programme aufmerksam geworden?
  - ○ Hast du auf die Geografie des Angebots geachtet? Z.B. nur regionale Programme/ deutschlandweite Programme?
  - ○ Hast du gezielt nach Programmen für Food Startups gesucht? Erfolgreich?
- Ausgeschlossen: Hast du Angebote oder Programme ausgeschlossen?
  - ○ Wusstest du von bestimmten Unterstützungsangeboten und hast diese aber bewusst nicht genutzt?
  - ○ Hast du Angebote abgelehnt, die dir angeboten wurden? Wenn ja, warum?
- Kooperation mit Akteuren: Mit welchen Akteuren hast du im Prozess zusammengearbeitet?
  - ○ Hast du dich während des gesamten Prozesses aktiv an einen der folgenden Akteure gewandt: Universitäten/ Forschungsinstitute, etablierte Unternehmen, staatliche Institutionen, Inkubatoren/Akzeleratoren, Investoren/VC/Business Angels? Wenn ja, welche, warum und wofür?

**Table A2.** *Cont.*

| | |
|---|---|
| Themenblock 5: Blindspots und Defizite im Förderungsangebot | • Blindspots: Hast du dich während des Prozesses gut unterstützt gefühlt?<br>  ○ Gab es Situationen, in denen du dir Unterstützung gewünscht hättest, aber nicht bekommen hast?<br>  ○ Insbesondere für dich als Lebensmittel Startup?<br>  ○ Auf einer Skala von 1–10 (1 das niedrigste,10 das höchste): wie würdest du die Unterstützung bewerten für Food Startups in Deutschland?<br>  ○ Weißt du von Unterstützungs- und Förderungsangeboten an anderen Orten, wo du dir wünschen würdest, dass diese auch in deiner Region angeboten würden?<br>• Angenommen du dürftest dir etwas wünschen: was würdest du machen, um das Förderungsangebot in Deutschland für Food Startups zu verbessern?<br>• Retrospektiv:<br>  ○ Gab es irgendeine Art von Unterstützung, die du dir gewünscht hättest? Irgendetwas, das du vermisst hast?<br>  ○ Gab es ein Programm, von dem du dir gewünscht hättest, dass du schon eher davon gewusst hättest? Wenn ja, warum und wie wärst du eher bzw. besser darauf aufmerksam geworden?<br>• Aktuelle Situation:<br>  ○ In welcher Phase befindest du dich gerade mit deinem Startup?<br>  ○ Welche Unterstützung benötigst du gerade? |
| Abschluss | • Gibt es noch etwas, was du gerne hinzufügen würdest oder vergessen hast, vorher zu sagen?<br>• Danke für deine Zeit. Gibt es noch etwas, das du von mir wissen möchtest? |
| | Aufnahme beenden |
| Verab-schiedung | • Weiteres Vorgehen (z.B. Erinnerung an Einverständniserklärung, wann Ergebnisse zu erwarten sind, Kontaktdetails bestätigen, etc.)<br>• Erneut bedanken<br>• Verabschieden |

*Appendix A.4. Interview Guide: Start-Ups (German, Short)*

Profil—Hintergrund-informationen des Startups

- Persönlicher Hintergrund: Erzähl mir doch etwas über dich als Person, deine Motivation für die Startup Gründung und deinen Hintergrund (Ausbildung, etc.)
- Gründungsgeschichte: als nächstes erzähl mir doch etwas über das Startup, die Entstehungsgeschichte und die Entwicklung? (Gründungsjahr, Mitgründer, Anzahl Gründer, Anzahl Personal derzeit, etc.)
- Food Startup: Identifizierst du dich als Food Startup? Wenn ja, wo entlang der Lieferkette platziert ihr euch? Wenn nein, inwieweit habt ihr mit Lebensmitteln zu tun?
- "Transformative Innovationen": Welches Problem willst du mit dem Startup lösen und denkst du, dass das einen signifikanten Einfluss auf das gesamte Lebensmittelsystem hat?
- Was glaubst du, welche Auswirkungen diese Innovation haben wird?

Herausforderungen

- Herausforderung: Welchen Herausforderungen standest du während der Gründung gegenüber?
- Phasen: Gab es unterschiedliche Herausforderung zu verschiedenen Zeitpunkten im Prozess, also z.B. in der Ideenphase und die Prototypen-Entwicklungsphase? Welche?
- Welche Ressourcen waren hilfreich während der verschiedenen Phasen des Startups?
  - → Beispiel: physisch, finanziell, Personal, immateriell (Patente, Trademark, Wissen, Daten)

Förderungsumgebung

- Unterstützung/ Förderung: Wurdest du während des (gesamten) Prozesses in irgendeiner Weise unterstützt? Wie?
- Zugang: Wie bist du auf die Angebote und Programme aufmerksam geworden?
- Ausgeschlossen: Hast du Angebote oder Programme ausgeschlossen? Warum?
- Kooperation mit Akteuren: Mit welchen Akteuren hast du im Prozess zusammengearbeitet?
- Hast du dich während des gesamten Prozesses aktiv an einen der folgenden Akteure gewandt: Universitäten/ Forschungsinstitute, etablierte Unternehmen, staatliche Institutionen, Inkubatoren/Akzeleratoren, Investoren/VC/Business Angels? Wenn ja, welche, warum und wofür?

Blindspots und Defizite im Förderungsangebot

- Blindspots: Hast du dich während des Prozesses gut unterstützt gefühlt?
- Retrospektiv: Gab es irgendeine Art von Unterstützung, die du dir gewünscht hättest? Irgendetwas, das du vermisst hast? Gab es ein Programm, von dem du dir gewünscht hättest, dass du schon eher davon gewusst hättest? Wenn ja, warum und wie wärst du eher bzw. besser darauf aufmerksam geworden?
- Angenommen du dürftest dir etwas wünschen: was würdest du machen, um das Förderungsangebot in Deutschland für Food Startups zu verbessern?
- Aktuelle Situation: In welcher Phase befindest du dich gerade mit deinem Startup?

**Appendix B. Interview Guide: Ecosystem Actors**

The interview guides for ecosystem actors were created in a comprehensive format with all sub-questions and a shorter version that was sent to the interviewees in advance.

*Appendix B.1. Interview Guide: Ecosystem Actors*

**Table A3.** Interview Guide: Ecosystem Actors.

| Einleitung | - Persönliche Vorstellung<br>　○　Für Teilnahme bedanken<br>　○　Vorstellung<br>- Vorstellung des Themas<br>　○　Ziel der Masterarbeit<br>　○　Fokus der Masterarbeit<br>　○　Definitionen<br>- Organisatorisches<br>　○　Wiederholen: Erlaubnis zur Aufzeichnung einholen<br>　○　Anonymität erklären<br>　○　Ergebnisse anbieten |
|---|---|
| | Aufnahme starten |
| Themenblock 1 (Narrativer Teil): Hintergrundinformationen | - Persönlicher Hintergrund: Erzähl mir doch etwas über dich als Person, deine Arbeit und was du mit Innovationen im Lebensmittelsystem und Food Startups zu tun hast?<br>　○　Was ist dein persönlicher Hintergrund? (Ausbildung)<br>　○　Inwieweit arbeitest du mit Innovationen zusammen?<br>　○　Inwieweit arbeitest du mit Food Startups zusammen?<br>　○　Inwieweit kennst du dich im Lebensmittelsystem aus?<br>　○　Was hast du mit Startups zu tun? |

**Table A3.** *Cont.*

| | |
|---|---|
| **Themenblock 2: Transformative Innovationen** | • Startups als Transformatoren: Inwieweit glaubst du, dass Startups im Vergleich zu etablierten Unternehmen Innovationen herausbringen?<br>○ Bringen Startups deiner Meinung nach disruptivere Innovationen heraus im Vergleich zu Unternehmen?<br>○ Was für Probleme werden von Startups prinzipiell in Angriff genommen bzw. gelöst? Gibt es einen Hauptbereich, mit denen sich Startups thematisch beschäftigen?<br>→ Beschäftigen sich Startups aktiv mit den großen Problemen im Lebensmittelsystem, wie beispielsweise Klimawandel, Lebensmittel-verschwendung, oder eher mit neuen Geschmacksrichtungen, etc.?<br>• "Transformative Innovationen": Inwieweit glaubst du, dass eine Veränderung des jetzigen Lebensmittelsystems vollzogen werden kann?<br>○ Können Innovationen eine Veränderung im Lebensmittelsystem hervorrufen?<br>○ Wer sind die Hauptakteure, um eine Veränderung hervorzurufen, Startups, Konsumenten, etablierte Unternehmen, etc.? |
| **Themenblock 3: Herausforderungen für Startups** | • Herausforderung: Was sind deiner Meinung nach die größten Herausforderungen, denen Startups, insbesondere Food Startups, gegenüber stehen?<br>○ Was waren die größten Herausforderungen, die du bei Startups beobachten kannst/konntest?<br>■ Produktentwicklung (Zusammensetzung Rezeptur, Prototypenentwicklung, Werkzeuge)<br>■ Richtlinien und Patente (Hygienestandards in Produktion, Patentstreit)<br>■ Kapital (insbesondere hohe Anfangsinvestitionen, Lohnproduktionen erst bei höherer Abnahme möglich)<br>■ Netzwerke & Beratung (Lebensmitteltechniker statt nur Business)<br>• Unterschiede: Gibt es Branchenspezifische Unterschiede bzw. Herausforderungen, insbesondere für Food Startups?<br>○ Unterscheiden sich die Herausforderungen für Food Startups im Vergleich zu anderen Startups?<br>○ Gibt es deiner Meinung nach spezifische Herausforderungen für Food-Startups? Wenn ja, welche?<br>→ Beispiele: Lebensmittelgesetze, Qualitätsmanagement, Patente, Marketing, Market Research, Kapital, etc.<br>○ Gibt es Unterschiede in den Herausforderungen, je nachdem, was für eine Innovation die Startups haben, z.B. Produkt, Prozess, Geschäftsmodell?<br>○ Gibt es Unterschiede, je nachdem, mit welchem Aspekt der Food Value Chain sich die Startups beschäftigen?<br>○ Findest du, dass Food-Startups ein spezielles Förderumfeld benötigen? Wenn ja, welches?<br>• Phasen: Gibt es unterschiedliche Herausforderung zu verschiedenen Zeitpunkten im Prozess, also z.B. in der Ideenphase und die Prototypen-Entwicklungsphase?<br>○ Welche Herausforderungen gibt es in den verschiedenen Phasen der Startups?<br>→ z.B. Ideenphase, pre-seed, seed, startup<br>○ Welche Resourcen sind hilfreich während der verschiedenen Phasen des Startups und inwieweit unterscheiden sie sich je nach Phase?<br>→ Beispiel: physisch, finanziell, Personal, immateriell (Patente, Trademark, Wissen, Daten)<br>• Kennst du Unterstützungsangebote, die du nicht befürwortest? |

**Table A3.** *Cont.*

| | |
|---|---|
| Themenblock 4: Förderungsumgebung | • Unterstützung/ Förderung: Welche Unterstützung und Förderunge wird von euch angeboten?<br>   ○ Gibt es spezielle Programme? Wenn ja, welche?<br>   ○ An wen richten sich die Angebote bzw. wer kann die Angebote wahrnehmen? → z.B. nur Studierende, regionaler Fokus, jeder<br>   ○ Gibt es unterschiedliche Angebote für die verschiedenen Startup Phasen?<br>   ○ Welches Angebot ist deiner Einschätzung nach das erfolgreichste?<br>• Zugang: Wie und wo macht ihr auf die Angebote und Programme aufmerksam?<br>   ○ Richten sich die Angebote primär an Teams aus der Region oder gelten sie deutschlandweit?<br>   ○ Woher kommen die Teams geografisch?<br>   ○ Wie erfahren die Teams von euren Angeboten?<br>• Zusätzliche Förderung: Gibt es noch weitere Angebote, die ihr aufbauen möchtet bzw. ergänzen möchtet?<br>   ○ Bist du mit eurem Angebot zufrieden? Wenn ja, warum?<br>   ○ Wie kann man deiner Meinung nach das Angebot noch verbessern?<br>   ○ Gibt es noch weitere Programme oder Angebote, die ihr auf- bzw. ausbauen möchtet? Welche und warum?<br>   ○ Gibt es Förderangebote, die deiner Meinung nach nicht erfolgreich sind bzw. im Vergleich weniger erfolgreich sind?<br>   ○ Werden Angebote von euch von den Teams auch schon mal abgelehnt? Wenn ja, welche und warum?<br>• Kooperation mit Akteuren: Mit welchen Akteuren arbeitet ihr zusammen?<br>   ○ Arbeitet ihr mit einen der folgenden Akteure zusammen bzw. sind Teil des Netzwerkes: Universitäten/ Forschungsinstitute, etablierte Unternehmen, staatliche Institutionen, Inkubatoren/Akzeleratoren, Investoren/VC/Business Angels? Wenn ja, welche, warum und wofür?<br>   ○ Welchen Akteur würdest du als besonders wichtig einstufen? (max. 2)<br>   ○ Inwieweit ist deiner Meinung nach die Kooperation verschiedener Akteure wichtig? |
| Themenblock 5: Blindspots und Defizite im Förderungsangebot | • Blindspots: Wie gut würdest du das Förderangebot in Deutschland für Innovationen und Startups, insbesondere Food-Startups einschätzen? Gerade auch im Vergleich zu Schweiz, Österreich und Luxemburg!<br>   ○ Gibt es Situationen, in denen du frustriert bist von dem aktuellen System der Förderung?<br>   ○ Insbesondere für die Förderung von Innovationen im Lebensmittelbereich?<br>   ○ Auf einer Skala von 1–10 (wo 1 das niedrigste ist und 10 das höchste): wie würdest du die Unterstützung bewerten für<br>      ■ Startups in Germany<br>      ■ Food Startups Germany<br>   ○ Weißt du von Unterstützungs- und Förderungsangeboten an anderen Orten, wo du dir wünschen würdest, dass sie auch in deiner Region angeboten würden?<br>• Wie würden sie die Unterstützungsangebote für Food Startups in Deutschland insgesamt bewerten?<br>• Was fehlt Ihrer Meinung nach an Förderangeboten in Deutschland bzw. kennst du Unterstützungsangebote für Food Startups, von dem du dir wünschst, dass es diese Unterstützung auch in Deutschland gibt?<br>• Angenommen du dürftest dir etwas wünschen: was würdest du machen, um das Förderungsangebot bzw. die gründungsfördernden Rahmenbedingungen in Deutschland für Food Startups zu verbessern? |
| Abschluss | • Gibt es noch etwas, was du gerne hinzufügen würdest oder vergessen hast, vorher zu sagen?<br>• Danke für deine Zeit. Gibt es noch etwas, das du von mir wissen möchtest?<br>• Ich beende dann jetzt die Aufnahme<br>• Weiteres Vorgehen |
| | Aufnahme beenden |
| Verab-schiedung | • Weiteres Vorgehen (z.B. Erinnerung an Einverständniserklärung, wann Ergebnisse zu erwarten sind, Kontaktdetails bestätigen, etc.)<br>• Erneut bedanken<br>• Verabschieden |

*Appendix B.2. Interview Guide: Ecosystem Actors (Short)*

Hintergrund

- Persönlicher Hintergrund: Erzähl mir doch etwas über dich als Person, deine Arbeit und was du mit Innovationen im Lebensmittelsystem und Food Startups zu tun hast?

Transformative Innovationen

- Start-ups als Transformatoren: Inwieweit glaubst du, dass Startups im Vergleich zu etablierten Unternehmen Innovationen herausbringen?
- "Transformative Innovationen" (Innovationen, die das System langfristig verändern, also radikale bzw. disruptive Innovationen): Inwieweit glaubst du, dass eine Veränderung des jetzigen Lebensmittelsystems vollzogen werden kann?
- Wer sind die Hauptakteure, um eine Veränderung hervorzurufen (z.B. Startups, Konsumenten, etablierte Unternehmen, etc.)?

Herausforderungen für Start-ups

- Herausforderung: Was sind deiner Meinung nach die größten Herausforderungen, denen Startups, insbesondere Food Startups, gegenüber stehen?
- Unterschiede: Gibt es Branchenspezifische Unterschiede bzw. Herausforderungen, insbesondere für Food Startups?
- Phasen: Gab es unterschiedliche Herausforderung zu verschiedenen Zeitpunkten im Prozess, also z.B. in der Ideenphase und die Prototypen-Entwicklungsphase?

Unterstützung/Förderung

- Welche Unterstützung und Förderung werden von euch angeboten? An wen?
  - → hier ist wie gesagt alles an Unterstützung gemeint, also auch Mentoring, Vernetzung, Wissen, etc.
- Zugang: Wie und wo macht ihr auf die Angebote und Programme aufmerksam?
- Zusätzliche Förderung: Gibt es noch weitere Angebote, die ihr aufbauen möchtet bzw. ergänzen möchtet?
- Kooperation mit Akteuren: Mit welchen Akteuren arbeitet ihr zusammen?
- Inwieweit ist deiner Meinung nach die Kooperation verschiedener Akteure wichtig?

Blindspots und Defizite im Förderungs-angebot

- Blindspots: Wie gut würdest du das Förderangebot in Deutschland für Innovationen und Startups, insbesondere Food-Start-ups einschätzen?
- Angenommen du dürftest dir etwas wünschen: was würdest du machen, um das Förderungsangebot bzw. die gründungsfördernden Rahmenbedingungen in Deutschland für Food Start-ups zu verbessern?

Abschluss

- Gibt es noch etwas, was du gerne hinzufügen würdest oder vergessen hast, vorher zu sagen?
- Weiteres Vorgehen

## Appendix C. Interview Guide: Food System Experts

The interview guides for ecosystem actors were created in a comprehensive format with all sub-questions and a shorter version that was sent to the interviewees in advance.

*Appendix C.1. Interview Guide: Food System Experts*

**Table A4.** Interview Guide: Food System Experts.

| | |
|---|---|
| **Einleitung** | • Persönliche Vorstellung<br>  ○ Für die Teilnahme bedanken<br>  ○ Vorstellung von mir als Person<br>• Vorstellung des Themas<br>  ○ Definitionen: Startup Support, Transformative Innovationen<br>• Organisatorisches<br>  ○ Erlaubnis zur Aufzeichnung einholen<br>  ○ Vertraulichkeit zusichern<br>  ○ Explizite Nachfrage: Anonymisieren oder nicht<br>  ○ Ergebnisse anbieten → Kontaktdaten |

<div align="center">Aufnahme starten</div>

| | |
|---|---|
| **Themenblock 1 (Narrativer Teil): Hintergrundinformationen** | • Persönlicher Hintergrund: Erzähl mir doch etwas über dich als Person, deine Arbeit und was du mit Innovationen im Lebensmittelsystem zu tun hast?<br>  ○ Was ist dein persönlicher Hintergrund? (Ausbildung)<br>  ○ Inwieweit arbeitest du mit Innovationen zusammen?<br>  ○ Inwieweit arbeitest du mit Food Startups zusammen?<br>  ○ Inwieweit kennst du dich im Lebensmittelsystem aus?<br>  ○ Was hast du mit Startups zu tun? |
| **Themenblock 2: Transformative Innovationen** | • "Transformative Innovationen": Inwieweit glaubst du, dass eine Veränderung des jetzigen Lebensmittelsystems vollzogen werden kann bzw. muss?<br>• Inwieweit spielen Innovationen bei dieser Transformation eine Rolle?<br>  ○ Können Innovationen eine Veränderung im Lebensmittelsystem hervorrufen?<br>• Wer sind die Hauptakteure, um eine Veränderung hervorzurufen, Startups, Konsumenten, etablierte Unternehmen, Forschung, etc.?<br>• Wie würdest du Innovationen im Lebensmittelbereich einteilen?<br>  ○ Z.B. nach Innovationstyp (Produkt/Prozess), entlang der Wertschöpfungskette, etc. |
| **Themenblock 3: Herausforderungen für Startups** | • Herausforderung: Was sind deiner Meinung nach die größten Herausforderungen, denen Food Innovationen, gegenüber stehen? Gerade aus deiner persönlichen Sicht.<br>  ○ Was waren die größten Herausforderungen, die du bei der Arbeit mit Lebensmitteln beobachten kannst/konntest?<br>    ■ Z. B. Produktentwicklung (Zusammensetzung Rezeptur, Prototypenentwicklung, Werkzeuge), Richtlinien und Patente (Hygienestandards in Produktion, Patentstreit), Kapital (insbesondere hohe Anfangsinvestitionen, Lohnproduktionen erst bei höherer Abnahme möglich), Netzwerke & Beratung (Lebensmitteltechniker statt nur Business)<br>• Unterschiede: Gibt es Branchenspezifische Besonderheiten bzw. Herausforderungen, insbesondere für Innovationen im Lebensmittelbereich bzw. insbesondere für Food Startups?<br>  ○ Unterscheiden sich die Herausforderungen für Lebensmittelinnovationen im Vergleich zu anderen Innovationen?<br>  ○ Gibt es deiner Meinung nach weitere spezifische Herausforderungen für Lebensmittel? Wenn ja, welche?<br>    → Beispiele: Lebensmittelgesetze, Qualitätsmanagement, Patente, Marketing, Market Research, Kapital, etc.<br>  ○ Gibt es Unterschiede in den Herausforderungen, je nachdem, was für eine Innovation die Startups haben, z.B. Produkt, Prozess, Geschäftsmodell?<br>  ○ Gibt es Unterschiede, je nachdem, mit welchem Aspekt der Food Value Chain sich die Startups beschäftigen?<br>  ○ Findest du, dass Food-Startups ein spezielles Förderumfeld benötigen? Wenn ja, welches? |

**Table A4.** *Cont.*

| Themenblock 4: Förderungsumgebung | • Unterstützung/ Förderung: Welche Unterstützung und Förderung ist für Innovationen (Produkte, Prozesse, etc.) entscheidend?<br>• Kooperation mit Akteuren: Mit welchen Akteuren arbeitet ihr zusammen?<br>   ○ Arbeitet ihr mit einen der folgenden Akteure zusammen bzw. sind Teil des Netzwerkes: Universitäten/ Forschungsinstitute, etablierte Unternehmen, staatliche Institutionen, Inkubatoren/Akzeleratoren, Investoren/VC/Business Angels? Wenn ja, welche, warum und wofür?<br>   ○ Welchen Akteur würdest du als besonders wichtig einstufen? (max. 2)<br>   ○ Inwieweit ist deiner Meinung nach die Kooperation verschiedener Akteure wichtig? |
|---|---|
| Themenblock 5: Blindspots und Defizite im Förderungsangebot | • Blindspots: Wie gut würdest du das Förderangebot in Deutschland für Innovationen und Startups, insbesondere Food-Startups einschätzen?<br>   ○ Gibt es Situationen, in denen du frustriert bist von dem aktuellen System der Förderung?<br>   ○ Insbesondere für die Förderung von Innovationen im Lebensmittelbereich?<br>   ○ Auf einer Skala von 1–10 (wo 1 das niedrigste ist und 10 das höchste): wie würdest du die Unterstützung bewerten für<br>     ■ Startups in Germany<br>     ■ Food Startups Germany<br>   ○ Weißt du von Unterstützungs- und Förderungsangeboten an anderen Orten, wo du dir wünschen würdest, dass sie auch in deiner Region angeboten würden?<br>• Wie würdest du die Unterstützungsangebote für Food Startups in Deutschland insgesamt bewerten?<br>• Was fehlt deiner Meinung nach an Förderangeboten in Deutschland bzw. kennst du Unterstützungsangebote für Food Startups, von dem du dir wünschst, dass es diese Unterstützung auch in Deutschland gibt?<br>• Angenommen du dürftest dir was wünschen: was würdest du machen, um das Förderungsangebot bzw. die gründungsfördernden Rahmenbedingungen in Deutschland für Food Startups zu verbessern? |
| Abschluss | • Gibt es noch etwas, was du gerne hinzufügen würdest oder vergessen hast, vorher zu sagen?<br>• Danke für deine Zeit. Gibt es noch etwas, das du von mir wissen möchten? |
| | Aufnahme beenden |
| Verab-schiedung | • Weiteres Vorgehen (z.B. Erinnerung an Einverständniserklärung, wann Ergebnisse zu erwarten sind, Kontaktdetails bestätigen, etc.)<br>• Erneut bedanken<br>• Verabschieden |

*Appendix C.2. Interview Guide: Food System Experts (Short)*

Profil—Hintergrund

- Persönlicher Hintergrund: Erzähl mir doch etwas über dich als Person, deine Arbeit und was du mit Innovationen im Lebensmittelsystem zu tun hast?

Transformative Innovationen

- "Transformative Innovationen": Inwieweit glaubst du, dass eine Veränderung des jetztigen Lebensmittelsystems vollzogen werden kann bzw. muss?
- Inwieweit spielen Innovationen bei dieser Transformation eine Rolle?
- Wer sind die Hauptakteure, um eine Veränderung hervorzurufen, Startups, Konsumenten, etablierte Unternehmen, Forschung, etc.?
- Wie würdest du Innovationen im Lebensmittelbereich einteilen?
  → z.B. nach Innovationstyp (Produkt/Prozess), entlang der Wertschöpfungskette, etc.

Herausforderungen für Innovationen im Lebensmittel-system

- Herausforderung: Was sind deiner Meinung nach die größten Herausforderungen, denen Food Innovationen, gegenüber stehen? Gerade aus deiner persönlichen Sicht.
- Was waren die größten Herausforderungen, die du bei der Arbeit mit Lebensmitteln / Innovationen beobachten kannst/konntest?

- Z. B. Produktentwicklung, Richtlinien und Patente, Kapital, Netzwerke & Beratung, etc.
- Unterschiede: Gibt es Branchenspezifische Besonderheiten bzw. Herausforderungen, insbesondere für Innovationen im Lebensmittelbereich bzw. insbesondere für Food Startups?
- Unterscheiden sich die Herausforderungen für Lebensmittelinnovationen im Vergleich zu anderen Innovationen?
- Findest du, dass Food-Startups ein spezielles Förderumfeld benötigen? Wenn ja, welches?

Förderungsumgebung

- Unterstützung/ Förderung: Welche Unterstützung und Förderung ist für Innovationen (Produkte, Prozesse, etc.) entscheidend?
- Kooperation mit Akteuren: Mit welchen Akteuren arbeitet ihr zusammen?
- Inwieweit ist deiner Meinung nach die Kooperation verschiedener Akteure wichtig?

Blindspots und Defizite im Förderungs-angebot

- Blindspots: Wie gut würdest du das Förderangebot in Deutschland für Innovationen und insbesondere Food-Startups einschätzen?
- Wie würdest du die Unterstützungsangebote für Food Startups in Deutschland insgesamt bewerten?
- Was fehlt deiner Meinung nach an Förderangeboten in Deutschland bzw. kennst du Unterstützungsangebote für Food Startups, von denen du dir wünschst, dass es diese Unterstützung auch (öfter) in Deutschland gibt?
- Angenommen du dürftest dir was wünschen: was würdest du machen, um das Förderungsangebot bzw. die gründungsfördernden Rahmenbedingungen in Deutschland für Food Startups zu verbessern?

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
