# Peer review of "Transforming the German Food System: How to Make Start-Ups Great!"

_sustainability, doi:10.3390/su14042363_

Round 1

Reviewer 1 Report

Dear author,

Your paper addresses an emerging issue in the area of innovation and the food sector. However, authors should make a remarkable effort to improve their manuscript to make it publishable. I would like to point out some aspects to improve your manuscript. I explain my concerns in more detail below. I ask authors to specifically address each of my comments in their response.

MAJOR COMMENTS

1) In the introduction, it is recommended to include the expected results (to captivate the reader). It would also be advisable for authors to include a paragraph describing the original contribution of their study and its contribution to the advancement of knowledge. Another aspect that should be included in the introduction is a paragraph justifying the interest of choosing Germany as the territorial unit of analysis: why choose Germany, what is the relevance of focusing the study on this country, what characteristics make Germany a territorial unit of interest for the study, and what is the relevance of focusing the study on this country, and what characteristics make Germany a territorial unit of interest for the study. Finally, it would be useful to include a last paragraph to briefly describe what the reader will find in the following sections.

2) In section 2 it is recommended that the authors make a notable effort to conduct a literature review. The review undertaken is insufficient. It is suggested that the authors incorporate several paragraphs to contextualise the challenges facing the agri-food sector. These challenges include the sustainable development of productive activity and digital transformation. Below, we suggest some papers currently published that may help them to complete the introduction:

Zhu, Z., Bai, Y., Dai, W., Liu, D., & Hu, Y. (2021). Quality of e-commerce agricultural products and the safety of the ecological environment of the origin based on 5G Internet of Things technology. Environmental Technology & Innovation22, 101462. https://doi.org/10.1016/j.eti.2021.101462

Jorge-Vázquez, J.; Chivite-Cebolla, M.P.; Salinas-Ramos, F. The Digitalization of the European Agri-Food Cooperative Sector. Determining Factors to Embrace Information and Communication Technologies. Agriculture 2021, 11, https://doi.org/10.3390/agriculture11060514

-Mozas-Moral, A., FERNÁNDEZ-UCLÉS, D., BERNAL-JURADO, E., & Medina-Viruel, M. J. (2016). Web quality as a determining factor in the online retailing of organic products in Spain. New Medit: Mediterranean Journal of Economics, Agriculture and Environment= Revue Méditerranéenne dʹEconomie Agriculture et Environment15(2), 28.

-Saetta, S., & Caldarelli, V. (2020). How to increase the sustainability of the agri-food supply chain through innovations in 4.0 perspective: A first case study analysis. Procedia Manufacturing42, 333-336. https://doi.org/10.1016/j.promfg.2020.02.083

- Jorge-Vázquez, J., Chivite-Cebolla, Mª.P. & Salinas-Ramos, F. (2019): “La transformación digital en el sector cooperativo agroalimentario español: situación y perspectivas”, CIRIEC-España, Revista de Economía Pública, Social y Cooperativa, 95, 39-70. DOI: 10.7203/CIRIEC-E.95.13002.

3) The description and justification of the methodology should be improved: why is this method important in this study, what are the advantages and disadvantages of this methodology, why apply this method and not another? Although the authors justify, in my opinion it is insufficient and should be completed from the perspective of other studies in the literature. It is also recommended to include in annexes the questionnaire used in the surveys carried out. It is recommended that the representativeness of the sample and the selection criteria be justified.

4) the authors must include a section discussing the results. Authors should discuss the results and how they can be interpreted from the per-spective of previous studies and of the working hypotheses. The findings and their implications should be discussed in the broadest context possible.

MINOR COMMENTS

1) It is recommended to specify the source of the figures and tables.

2) It is recommended to present the graphs in colour to facilitate their visualisation and interpretation.

3) There is a problem in the text with the references: "+ Error! Reference source not found".

4) REFERENCES: references should conform to the style standards proposed by the journal:

References must be numbered in order of appearance in the text (including citations in tables and legends) and listed individually at the end of the manuscript. We recommend preparing the references with a bibliography software package, such as EndNote, ReferenceManager or Zotero to avoid typing mistakes and duplicated references. Include the digital object identifier (DOI) for all references where available.

Citations and references in the Supplementary Materials are permitted provided that they also appear in the reference list here.

In the text, reference numbers should be placed in square brackets [ ] and placed before the punctuation; for example [1], [1–3] or [1,3]. For embedded citations in the text with pagination, use both parentheses and brackets to indicate the reference number and page numbers; for example [5] (p. 10), or [6] (pp. 101–105).

  1. Author 1, A.B.; Author 2, C.D. Title of the article. Abbreviated Journal Name Year, Volume, page range.
  2. Author 1, A.; Author 2, B. Title of the chapter. In Book Title, 2nd ed.; Editor 1, A., Editor 2, B., Eds.; Publisher: Publisher Location, Country, 2007; Volume 3, pp. 154–196.
  3. Author 1, A.; Author 2, B. Book Title, 3rd ed.; Publisher: Publisher Location, Country, 2008; pp. 154–196.
  4. Author 1, A.B.; Author 2, C. Title of Unpublished Work. Abbreviated Journal Name stage of publication (under review; accepted; in press).
  5. Author 1, A.B. (University, City, State, Country); Author 2, C. (Institute, City, State, Country). Personal communication, 2012.
  6. Author 1, A.B.; Author 2, C.D.; Author 3, E.F. Title of Presentation. In Title of the Collected Work (if available), Proceedings of the Name of the Conference, Location of Conference, Country, Date of Conference; Editor 1, Editor 2, Eds. (if available); Publisher: City, Country, Year (if available); Abstract Number (optional), Pagination (optional).
  7. Author 1, A.B. Title of Thesis. Level of Thesis, Degree-Granting University, Location of University, Date of Completion.
  8. Title of Site. Available online: URL (accessed on Day Month Year).

I hope these comments will be helpful. Good luck with publishing!

Best regards,

Author Response

Dear Sir/Madam,

Thank you for reviewing our contribution. We have added our changes and comments and hope to fulfil now all requirements for publication.

Best regards,

The authors

Reviewer 2 Report

This paper deals with innovation from the perspective of start-ups in Germany which can change or transform the current food system. The paper has a good structure and the flow is very clear. The methodology section is presented tight but enough. But some issues have to be Improved:

  1. I have not found any connection to the topic of the journal “Sustainability”. Exception “United Nations sustainable development goals” which is also looking very poor. Add more connection with sustainability in the literary review and conclusions section would be more favourable to journal policy
  1. Pages 8-10. Have fail references “Error! Reference source not found.”
  1. Other brackets should be used for references. It’s been mixing In text with other none-references like on page 3, line 16-18.
  2. Page 3. There is wrong order of table appearance. Please check it. The table should be after it is first mentioned.
  3. The author uses different methodology: players (abstract) – actors (body), It should be presented one way
  4. Table 6. Check this category “Most/most minor important actor” Most/most? This Table has no header
  5. Figure 1 is all grey. Nothing can be understood.
  6. Authors use the concept  “innovation ecosystem” It’s not clearly understood what the authors mean by it. Why is it innovative?

Generally, the paper is very good presented.  One more doubt is a small number of interviewers. According to the title, I understand this is some preliminary study. I hope the Authors will made more research in future and extend their number. This information could mention in conclusions.

Author Response

(The authors gave the same response as above.)

Reviewer 3 Report

The paper brings results of the study focused on the improvement of starts-ups in the German food system. The structure of the paper follows a structure of a scientific paper, its improvement is recommended. However, the relationship to sustainability aspects is unclear and therefore it is doubtful if the paper suits the scope of the journal Sustainability.

My comments for improvement of the paper are as follows:

  1. Give information regarding the relationship of the paper to sustainability
  2. Introduction: add information about similar research works, extend literature review and better underline the novelty of the paper.
  3. Methods used by analysing and processing research data should be described more precise.
  4. Discussion of results with other works is missing - it should be added.
  5. Conclusions:
  • Highlight the contribution of the paper to development of scientific knowledge
  • Give limitation of the study/research presented in the paper
  • Give information to directions of the future research

Author Response

(The authors gave the same response as above.)

Round 2

Reviewer 1 Report

Dear authors,

I think the authors have made a remarkable effort to improve the article. They have improved the structure and organisation of the content. They have also made an effort to expand the review of the research background and to incorporate more recent references. However, they have not included a discussion of the results. In my opinion the discussion of results is fundamental to the article. 

  1. The bibliography needs to be standardised. The rules of the Journal are not followed.
  2. The bibliographic reference (34) is incorrect. According to the authors, it refers to this article:

Jorge-Vázquez, J.; Chivite-Cebolla, M.P.; Salinas-Ramos, F. The Digitalization of the European Agri-Food Cooperative Sector. Determining Factors to Embrace Information and Communication Technologies. Agriculture 2021, 11, https://doi.org/10.3390/agriculture11060514

  1. It is important to include the DOI.

Wish you all the best. Best regards,

Author Response

Dear Reviewer thank you very much for reviewing our contribution again.

We now extended the discussion and hope to fulfill your expectations.

Best regards, 

The authors.

Comments and made changes

I think the authors have made a remarkable effort to improve the article. They have improved the structure and organisation of the content. They have also made an effort to expand the review of the research background and to incorporate more recent references.  However, they have not included a discussion of the results. In my opinion the discussion of results is fundamental to the article. 

We now added the section 2.3 Reasons of Failure for Start-ups (see lines 243-270) for comparing our findings against findings from the literature. Furthermore, we arranged the conclusion chapter and renamed it into “Discussion and Conclusions”. Furthermore, we subdivided this chapter into three section. Section 5.1 (5.1 Challenges of Food Start-up in the Food System) was written completely new (lines 882-917) whereas sections 5.2 and 5.3 came mainly from the existing “Conclusion”. Nonetheless, these have been rearranged and references to the existing research has been made.

  1. The bibliography needs to be standardised. The rules of the Journal are not followed.

From the sustainability website we got the information that Sustainability now accepts free format submission, provided that we use the consistent formatting throughout. This is done now in text via the bracket and the consistent application of the chosen IEEE style in the applied reference manager Mendeley. If the editor want a different format we would adapt and modify.

  1. The bibliographic reference (34) is incorrect. According to the authors, it refers to this article:

Thank you for this hint. We corrected this. Before we refererred to the wrong publication of Jorge-Vázquez.

Jorge-Vázquez, J.; Chivite-Cebolla, M.P.; Salinas-Ramos, F. The Digitalization of the European Agri-Food Cooperative Sector. Determining Factors to Embrace Information and Communication Technologies. Agriculture 2021, 11, https://doi.org/10.3390/agriculture11060514

  1. It is important to include the DOI.

Where existent a DOI was provided.

Reviewer 3 Report

The paper was revised according to almost all my comments. However, discussion is still missing - it should be improved.

Author Response

Dear Reviewer thank you very much for reviewing our contribution again.

We now extended the discussion and hope to fulfill your expectations.

Best regards,

The authors.

We now added the section 2.3 Reasons of Failure for Start-ups (see lines 243-270) for comparing our findings against findings from the literature. Furthermore, we arranged the conclusion chapter and renamed it into “Discussion and Conclusions”. Furthermore, we subdivided this chapter into three section. Section 5.1 (5.1 Challenges of Food Start-up in the Food System) was written completely new (lines 882-917) whereas section 5.2 and 5.3 came mainly from the existing “Conclusion”. Nonetheless, these have been rearranged and references to the existing research has been made.